# Instance-Prototype Affinity Learning for Non-Exemplar Continual Graph Learning

## Abstract

Graph Neural Networks (GNN) endure catastrophic forgetting, undermining their capacity to preserve previously acquired knowledge amid the assimilation of novel information. Rehearsal-based techniques revisit historical examples, adopted as a principal strategy to alleviate this phenomenon. However, memory explosion and privacy infringements impose significant constraints on their utility. Non-Exemplar methods circumvent the prior issues through Prototype Replay (PR), yet feature drift presents new challenges. In this paper, our empirical findings reveal that Prototype Contrastive Learning (PCL) exhibits less pronounced drift than conventional PR. Drawing upon PCL, we propose **I**nstance-**P**rototype **A**ffinity **L**earning (**IPAL**), a novel paradigm for Non-Exemplar Continual Graph Learning (NECGL). Exploiting graph structural information, we formulate Topology-Integrated Gaussian Prototypes (TIGP), guiding feature distributions towards high-impact nodes to augment the model's capacity for assimilating new knowledge. Instance-Prototype Affinity Distillation (IPAD) safeguards task memory by regularizing discontinuities in class relationships. Moreover, we embed a Decision Boundary Perception (DBP) mechanism within PCL, fostering greater inter-class discriminability. Evaluations on four node classification benchmark datasets demonstrate that our method outperforms existing state-of-the-art methods, achieving a better trade-off between plasticity and stability.

## 1 Introduction

As a potent paradigm for graph data analysis, Graph Neural Networks (GNN) Hamilton et al. (2017); Kipf & Welling (2016); Veličković et al. (2017); Xu et al. (2018) have garnered significant academic attention in recent years. However, most existing studies Bi et al. (2023); Dong et al. (2022); Wu et al. (2019) adhere to a static data regime, where the complete training set is available upfront and model parameters remain immutable after initial optimization. The real world exhibits an intrinsically dynamic nature, with information incessantly generated, such as user interactions on social media or the dissemination of domain-specific publications, posing considerable exigencies for static modeling paradigms. Continual Graph Learning (CGL) endeavors to assimilate novel knowledge while retaining previously acquired representations. Nonetheless, distributional disparities across tasks frequently precipitate catastrophic forgetting, manifesting as marked performance degradation on earlier tasks. Rehearsal-based approaches Arani et al. (2022); Zhou & Cao (2021) ameliorate this challenge by retrospectively incorporating a curated subset of exemplars from prior tasks. However, since representative exemplars are retained in the memory buffer for each task, the memory footprint escalates with longer task sequences, potentially culminating in memory explosion. Moreover, in privacy-sensitive contexts, access to raw examples may be constrained.

To surmount the challenges identified above, we examine Non-Exemplar Continual Graph Learning (NECGL), a more stringent paradigm that prohibits access to prior raw examples when encountering new tasks. In this context, catastrophic forgetting is further exacerbated. Existing Non-Exemplar approaches Li et al. (2024); Magistri et al. (2024); Ren et al. (2023); Wang et al. (2023) revisit historical prototype representations—the embeddings of prior classes in the encoder's latent space—to overcome this limitation via Prototype Replay (PR). However, successive updates to model parameters render earlier class prototypes progressively obsolete, giving rise to feature drift—reflected in their misalignment within the evolving feature space. Most existing approaches rectify prior class prototypes through feature drift compensation tech-

niques Gomez-Villa et al. (2024); Yu et al. (2020) post-training, or apply knowledge distillation Hinton et al. (2015); Zhu et al. (2021) online to regulate the evolution of feature space. While feature drift remains an inherent challenge in NECGL, empirical evidence suggests that Prototype Contrastive Learning (PCL) Li et al. (2025) induces less drift than conventional PR.

As shown in Figure 1, we visualize the extent of feature drift on base task samples across the CS-CL and CoraFull-CL datasets. Conventional PR formulates training with a cross-entropy objective, drawing on prior class prototypes for classification. As model parameters evolve, prototype misalignment intensifies, resulting in greater prediction errors and exacerbated feature drift. PCL encourages learning relational structures between instances and prototypes, explicitly distinguishing previously encountered classes from novel ones, thus attenuating task interference. *To elucidate this phenomenon, we introduce Kullback–Leibler (KL) divergence to quantify feature drift and provide a rigorous theoretical analysis and proof in Appendix A.* In con-

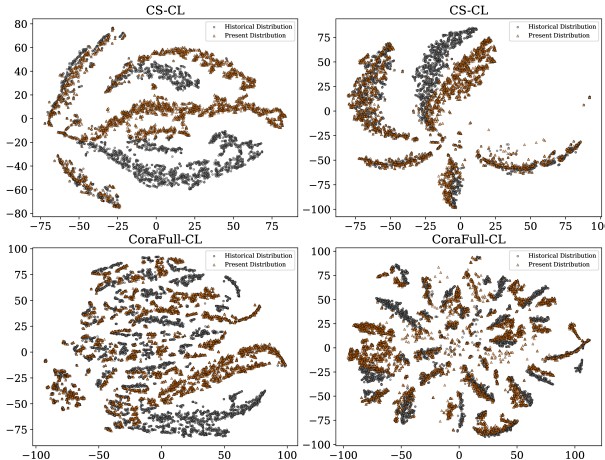

Figure 1: Visualization of feature drift for conventional Prototype Replay (left column) and Prototype Contrastive Learning (right column) on the CS-CL and CoraFull-CL datasets.

trast to Graph Contrastive Learning Hassani & Khasahmadi (2020); Xu et al. (2021), PCL inherently establishes positive pairs between instances and class-aligned prototypes, and negative pairs with inter-class prototypes, obviating the need for predefined augmentation strategies such as edge perturbation or attribute masking. Motivated by this observation, we propose a novel NECGL paradigm built upon PCL.

While existing Non-Exemplar methods sustain model stability through PR and knowledge distillation, notable limitations persist. First, most existing works Cheng et al. (2024); Li et al. (2024; 2025); Magistri et al. (2024); Ren et al. (2023) derive prototypes by averaging feature representations, *yet this isotropic way neglects the varying importance of nodes*. For graph-structured data, node significance is shaped by unique topologies and neighbor influences. Second, feature distillation is extensively applied in Non-Exemplar methods due to its inherent plug-and-play functionality. Nonetheless, recent research Magistri et al. (2024) indicates that *it imposes excessive constraints on the feature space*, inhibiting model plasticity. Third, a solitary prototype is often *insufficient to capture the complete distribution of a class*. In the context of PCL, the exclusive reliance on a single class prototype can give rise to inter-class ambiguity, thereby exacerbating catastrophic forgetting.

To overcome the above limitations, we propose **I**nstance-**P**rototype **A**ffinity **L**earning (**IPAL**), a novel framework tailored for NECGL. We evaluate node impact via the PageRank algorithm Page et al. (1999) and generate Topology-Integrated Gaussian Prototypes (TIGP), directing class distributions towards high-impact nodes to facilitate the assimilation of new knowledge. To combat catastrophic forgetting, we propose Instance-Prototype Affinity Distillation (IPAD), aligning instance-prototype relationships for more flexible regularization of the feature space. Notably, IPAD seamlessly integrates with PCL, providing distinct advantages over feature distillation. Moreover, we embed Decision Boundary Perception (DBP) mechanism into PCL to promote sharper inter-class delineation by repelling instances proximal to decision boundaries.

***Contributions.*** The main contributions of this paper are as follows: i) We propose IPAL, a novel paradigm tailored for NECGL that strikes a favorable trade-off between stability and plasticity; ii) We utilize the PageRank algorithm to generate more robust TIGP, integrating graph topology into the prototype computation to amplify learning capacity; iii) We design IPAD, a knowledge distillation method inherently compatible with PCL, enabling more flexible retention of prior knowledge; iv) We incorporate the DBP mechanism into the PCL objective for clearer inter-class separation; v) Extensive experiments on four node classification benchmark datasets demonstrate that our proposed IPAL outperforms existing state-of-the-art methods in the Non-Exemplar scenario.

## 2 RELATED WORK

### 2.1 CONTINUAL GRAPH LEARNING

CGL seeks to assimilate new knowledge while preventing GNN from forgetting historical knowledge. Prior studies adopted regularization, rehearsal, parameter isolation, or their combinations to mitigate catastrophic forgetting. Regularization-based methods Aljundi et al. (2018); Kirkpatrick et al. (2017); Li & Hoiem (2017); Liu et al. (2021) reinforce constraints on pivotal parameters by quantifying their significance, or facilitate output alignment via knowledge distillation applied to the model's logits. Rehearsal-based methods Liu et al. (2023); Zhang et al. (2022b); Zhou & Cao (2021) store task-specific exemplars in a memory buffer for replay when learning new tasks. Parameter isolation methods Niu et al. (2024); Zhang et al. (2023) prevent inter-task interference by assigning separate parameters or learning task-specific submodules. This work focuses on rehearsal-based methods for their superior performance and closer resemblance to human learning. Despite their efficacy, prolonged task sequences can impose substantial memory burdens, and privacy restrictions may limit access to historical data.

### 2.2 NON-EXEMPLAR CONTINUAL LEARNING

Non-Exemplar Continual Learning (NECL) updates models without revisiting prior raw examples. Existing studies Li et al. (2024); Magistri et al. (2024); Ren et al. (2023); Wang et al. (2023) circumvent memory and privacy concerns via PR, rather than replaying raw examples. However, feature drift remains a fundamental flaw of these approaches. While feature distillation curbs substantial variations in the feature space, Magistri et al. (2024) argued it induces excessive regularization, yielding performance akin to freezing the backbone after the base task Petit et al. (2023). To reconcile historical prototypes with the new feature space, Li et al. (2025); Magistri et al. (2024); Wang et al. (2023); Yu et al. (2020); Zhai et al. (2024) quantified prototype drift via statistical measures derived from new task samples, while Cheng et al. (2024); Gomez-Villa et al. (2024); Li et al. (2024) harnessed learnable neural networks for adaptive compensation. In this paper, we empirically observe from Figure 1 that PCL exhibits less prototype drift than conventional PR trained with cross-entropy, which motivates the proposal of IPAL.

## 3 PRELIMINARIES

### 3.1 PROBLEM FORMULATION

This paper explores the Class-Incremental Learning (CIL) setting, wherein a GNN is optimized consecutively over a task sequence $\mathcal{T} = \{\mathcal{T}_0, \mathcal{T}_1, ..., \mathcal{T}_N\}$ with $|\mathcal{T}| = N+1$. Each task $\mathcal{T}_{t \leq N} = \{\mathcal{G}_t, \mathcal{Y}_t\}$ is defined as a semi-supervised node classification task, where $\mathcal{G}_t = \{\mathcal{V}_t, \mathcal{E}_t\}$ denotes the task graph with node set $\mathcal{V}_t$ and edge set $\mathcal{E}_t$. $\mathcal{E}_t$ can be expressed through a binary adjacency matrix $\mathbf{A}_t$, where 1 indicates an edge and 0 its absence. The label set is given by $\mathcal{Y}_t = \{y_t^1, y_t^2, ..., y_t^{c_t}\}$, with $\mathcal{Y}_i \cap \mathcal{Y}_j = \emptyset$ for $i \neq j$. In the NECGL paradigm, the base task $\mathcal{T}_0$ typically comprises considerably more data than each incremental task $\mathcal{T}_t$ $(t > 0)$, facilitating GNN pretraining for improved incremental adaptation, i.e., $c_0 \gg c_t$. Crucially, when learning new tasks, access to data from prior tasks is rigorously restricted, allowing only the current task's data. Our aim is to train a GNN on the task sequence $\mathcal{T}$ to attain superior performance across all previously encountered tasks. For a model with an encoder $\mathcal{F}_{\theta_t}(\cdot)$ and a linear classifier $g_{\phi_t}(\cdot)$, existing approaches predominantly employ PR to alleviate catastrophic forgetting, with the optimization objective formalized as follows:

$$\mathcal{L}_{PR} = \mathbb{E}_{(x_t, y_t) \in \mathcal{T}_t} \left[ -y_t^c \log g_{\phi_t}(\mathcal{F}_{\theta_t}(x_t^c)) \right] + \mathbb{E}_{(f_{\mathcal{M}}^m, y_{\mathcal{M}}^m) \in \mathcal{M}} \left[ -y_{\mathcal{M}}^m \log g_{\phi_t}(f_{\mathcal{M}}^m) \right], \quad (1)$$

where $\mathcal{M}$ denotes the memory buffer storing historical class prototypes (i.e., Gaussian distributions with mean $\mu_m$ and diagonal covariance $\sigma_m^2$), from which feature representations $f_{\mathcal{M}}^m$ reflecting past task distributions are sampled for replay during new task learning.

### 3.2 PROTOTYPE CONTRASTIVE LEARNING

PCL, rooted in Prototypical Networks Snell et al. (2017) from few-shot learning, captures the semantic association between instance-wise and class-wise representations. The optimization objective is

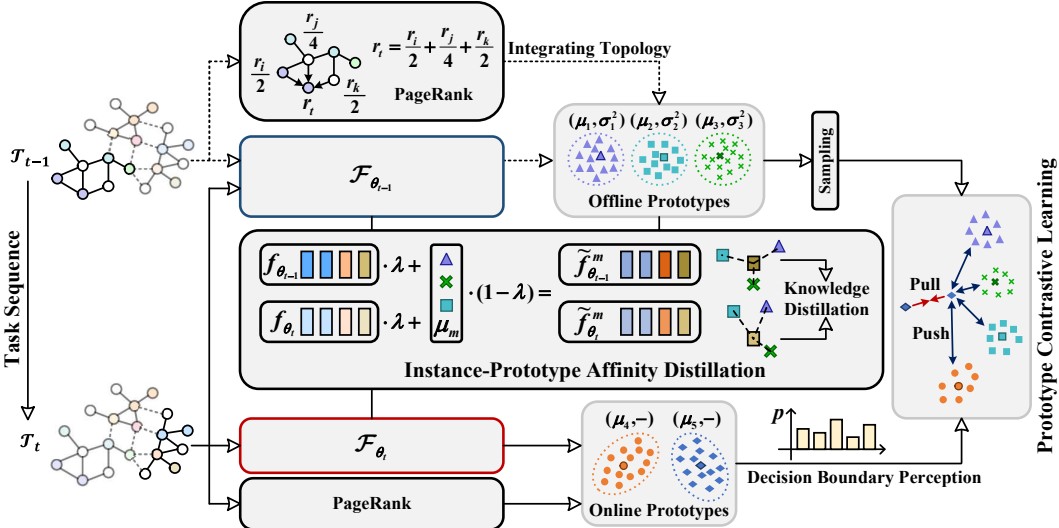

Figure 2: The overall pipeline of the proposed IPAL framework. Upon the culmination of task $\mathcal{T}_{t-1}$, the TIGP are derived and offline archived in memory buffer $\mathcal{M}$. Following the onset of task $\mathcal{T}_t$, online prototypes are dynamically updated and integrated with offline prototypes for PCL. In this regard, IPAD safeguards prior task memory via relational distillation, while DBP ensures the clear demarcation of newly encountered classes. Best viewed in color.

formally stated as follows:

$$\mathcal{L}_{PCL} = \mathbb{E}_{(x_t, y_t) \in \mathcal{T}_t} \left[ -\log \frac{e^{\mathcal{F}_{\theta_t}(x_t^c)^\top \cdot \mu_c / \tau}}{e^{\mathcal{F}_{\theta_t}(x_t^c)^\top \cdot \mu_c / \tau} + \sum_{j \neq c} e^{\mathcal{F}_{\theta_t}(x_t^c)^\top \cdot \mu_j / \tau}} \right], \tag{2}$$

where $\mu_{k \in \mathcal{Y}_{0:t}} = \frac{\sum_{(x,y) \in \mathcal{T}} \mathbb{1}\{y=k\} \mathcal{F}_\theta(x)}{\sum_{(x,y) \in \mathcal{T}} \mathbb{1}\{y=k\}}$ denotes the prototype for the class $k$, and $\tau$ is the temperature scaling factor that controls the distributional smoothness. In this paper, offline prototypes from prior tasks in $\mathcal{M}$ and current task instances form negative pairs, while online prototypes and label-matching instances from the current task form positive pairs. Set apart from instance-level contrastive learning (i.e., with anchors and positives/negatives as individual instances), Eq. 2 achieves competitive performance with minimal computational overhead by considering only interactions between each instance and a limited set of class prototypes. Relative to conventional PR, PCL confers superior resistance to feature drift (demonstrated in Section 1). The above merits justify utilizing PCL for NECGL.

## 4 METHODOLOGY

Figure 2 outlines the overall pipeline of the proposed IPAL. Founded upon PCL, we devise three pivotal components—Topology-Integrated Gaussian Prototypes (TIGP), Instance-Prototype Affinity Distillation (IPAD), and Decision Boundary Perception (DBP)—each meticulously tailored to rectify the three intrinsic limitations of existing Non-Exemplar methods articulated in Section 1. In the following, we provide a detailed analysis of each component.

### 4.1 TOPOLOGY-INTEGRATED GAUSSIAN PROTOTYPES

NECGL prohibits access to raw examples from prior tasks. To combat the intensified catastrophic forgetting, existing Non-Exemplar methods model each encountered class $k$ with a Gaussian distribution $\mathcal{N}(\mu_k, \sigma_k^2)$ after training, and retain it as a class prototype in the memory buffer $\mathcal{M}$. Although somewhat effective, treating all nodes uniformly in graph-structured data is untenable. Owing to the distinctiveness of graph topology, nodes within disparate neighborhoods exert differential influence on their adjacent counterparts. For example, prominent celebrities often possess a vast following, and their actions tend to wield greater influence on society. Indeed, such nodes tend to be more

indicative than low-degree ones. We steer the model to align class distributions with high-impact nodes, aiding the assimilation of new knowledge. Drawing inspiration from Page et al. (1999), node importance is evaluated via the PageRank algorithm, formalized as follows:

$$r_t = \alpha \sum_{j \in \mathcal{N}^{in}(t)} \frac{r_j}{d_j^+} + (1 - \alpha),$$
(3)

where $\mathcal{N}^{in}(t)$ denotes the set of incoming neighbors of node $t$, and $d_j^+ = |\mathcal{N}^{out}(j)|$ represents the out-degree of node $j$. $r_t$ is the PageRank for node $t$, and $\alpha$ is the damping factor. We then reweight the node contributions with the aid of PageRank to compute the TIGP for each class as follows:

$$\mu_k = \frac{\sum_{(x,y) \in \mathcal{T}} r_x \mathbb{1}\{y = k\} \mathcal{F}_\theta(x)}{\sum_{(x,y) \in \mathcal{T}} r_x \mathbb{1}\{y = k\}}, \quad \sigma_k^2 = \mathrm{diag}\left( \frac{\sum_{(x,y) \in \mathcal{T}} r_x \mathbb{1}\{y = k\} (\mathcal{F}_\theta(x) - \mu_k)^2}{\sum_{(x,y) \in \mathcal{T}} r_x \mathbb{1}\{y = k\}} \right).$$
(4)

Importantly, PageRank is computed once per task, avoiding recalculation in later iterations and imposing no extra burden on training. Moreover, offline prototypes are derived for all classes at the conclusion of each task, while dynamic online prototypes are instantiated for emerging classes throughout the PCL process to promote the integration of new knowledge. We visualize the performance heatmaps for mean-based prototypes and TIGP on three benchmark datasets. Figure 3 demonstrates that TIGP markedly surpasses mean-based prototypes, yielding notably enhanced plasticity.

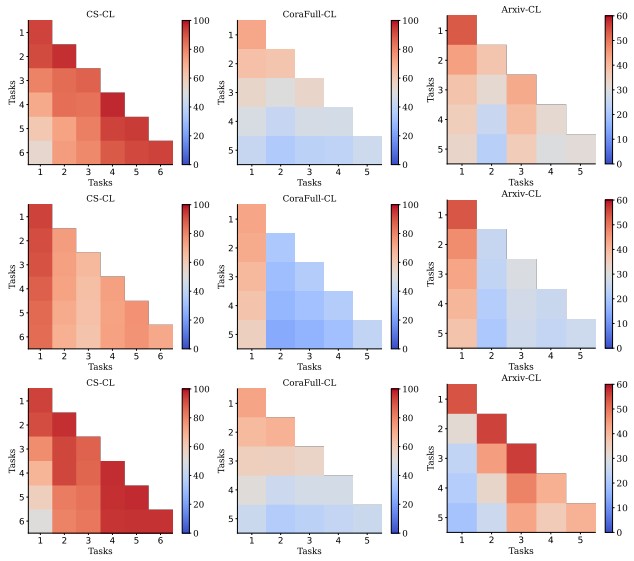

### 4.2 INSTANCE-PROTOTYPE AFFINITY DISTILLATION

A fundamental flaw of NECGL is feature drift (refer to Figure 1), wherein continual updates to the GNN render previously stored prototypes in memory buffer $\mathcal{M}$ increas-

Figure 3: Performance heatmaps on three datasets are shown. Top row: Mean-based prototypes for PCL; Middle row: Feature distillation to mitigate feature drift; Bottom row: Our proposed IPAL, which integrates TIGP and IPAD to better balance the trade-off between plasticity and stability.

ingly obsolete and incompatible with the evolving feature space, thereby exacerbating the risk of catastrophic forgetting during replay. A direct yet effective strategy involves replaying historical prototypes to the classifier $g_{\phi_t}(\cdot)$, while regularizing the current parameters $\theta_t$ toward their previous optimum $\theta_{t-1}$. A prevalent paradigm in constraining $\theta_t$ is feature distillation, formally defined as follows:

$$\mathcal{L}_{FD} = \mathbb{E}_{(x_t, y_t) \in \mathcal{T}_t} \|\mathcal{F}_{\theta_t}(x_t) - \mathcal{F}_{\theta_{t-1}}(x_t)\|_2.$$
(5)

Only the data from the current task $\mathcal{T}_t$ is permissible for use. While it substantially alleviates the drift issue, we find that feature distillation may sacrifice model plasticity for greater stability. As depicted in the middle row of Figure 3, feature distillation compels the model to predominantly preserve the initial feature space, with incremental learning merely embedding new class distributions within this confined space, thus impeding the assimilation of novel knowledge. Our observation concurs with Magistri et al. (2024), which reveals that feature distillation may degrade continual learning into merely fine-tuning the classifier while freezing the backbone after the base task.

To circumvent the rigidity of feature distillation, we resort to IPAD, which upholds intra- and inter-class instance–prototype relations for more flexible regularization. Moreover, it exhibits intrinsic compatibility with the PCL objective function, enabling more reliable control. To elaborate, the embeddings $f_{\theta_{t-1}}$ and $f_{\theta_t}$ for the current task samples $(x_t, y_t) \in \mathcal{T}_t$ are derived from the preceding and current GNNs, $\mathcal{F}_{\theta_{t-1}}(\cdot)$ and $\mathcal{F}_{\theta_t}(\cdot)$, respectively. Next, we utilize the Mixup strategy Wang

et al. (2021); Zhang et al. (2017) to interpolate them with prior class prototypes, synthesizing virtual features that maintain close proximity to the prototypes, thus ensuring clear class delineation. The formalized expression is as follows:

$$\tilde{f}_{\theta_{t-1}}^m = \lambda f_{\theta_{t-1}} + (1-\lambda)\mu_m, \quad \tilde{f}_{\theta_t}^m = \lambda f_{\theta_t} + (1-\lambda)\mu_m, \qquad (6)$$

where $\lambda \in [0, 0.4]$ is drawn from a Beta distribution, i.e., $\lambda \sim \mathrm{Beta}(9, 21)$. For efficiency, only a subset of $\mathcal{T}_t$ is engaged in Mixup. To prevent semantic noise in synthetic features, we further enforce pseudo-label filtration for label consistency:

$$\mathbf{idx} = \arg\max_{m \in \mathcal{Y}_{0:t-1}} \left( \tilde{f}_{\theta_{t-1}}^{m\top} \cdot \mu_m \right) == y_m, \qquad (7)$$

where $\mathbf{idx}$ denotes a boolean tensor that selects the synthetic features $\tilde{f}_{\theta_{t-1}}^m[\mathbf{idx}]$ and $\tilde{f}_{\theta_t}^m[\mathbf{idx}]$ whose labels match those of the prototypes. Ultimately, these retrieved synthetic features are exploited to enable effective IPAD, formulated as follows:

$$\mathcal{L}_{AD} = \mathbb{E}_{(x_t, y_t) \in \mathcal{S}_t} \left[ \sum_{m \in \mathcal{Y}_{0:t-1}} \| \tilde{f}_{\theta_t}^m[\mathbf{idx}]^\top \cdot \mu_m - \tilde{f}_{\theta_{t-1}}^m[\mathbf{idx}]^\top \cdot \mu_m \|_2 \right], \qquad (8)$$

where $\mathcal{S}_t \subseteq \mathcal{T}_t$, subset sampling, particularly on large-scale datasets, optimizes computational efficiency while regularizing model parameters to resist feature drift and catastrophic forgetting.

### 4.3 DECISION BOUNDARY PERCEPTION

PCL capitalizes on instance-prototype relationships for incremental learning, minimizing the distance between instances and prototypes with congruent labels, while maximizing the separation between those with disparate labels. Prototypes are typically assumed to be sufficiently representative class-level features. However, in practice, class distributions may exhibit considerable diversity, rendering a single prototype inadequate for comprehensive representation. When generalized to new tasks, this can give rise to distributional entanglement across classes, resulting in inter-class ambiguity.

In this paper, we augment PCL with a DBP mechanism, which facilitates explicit inter-class disentanglement by capturing relational dynamics with boundary-adjacent instances. Information entropy Shannon (1953; 1948) is harnessed to quantify the inherent predictive uncertainty of each node. High-entropy nodes, often residing near decision boundaries as hard instances, can be leveraged to promote sharper inter-class separation. Given the embedding $f_{\theta_t}^c$ of a node from task $\mathcal{T}_t$ with class $c$, its information entropy is calculated as follows:

$$p(x_t^c) = softmax(f_{\theta_t}^{c\top} \cdot [\mu_k]_{k \in \mathcal{Y}_{0:t}}), \quad \mathcal{H}(x_t^c) = -\sum_{k \in \mathcal{Y}_{0:t}} p(x_t^c) \log p(x_t^c), \qquad (9)$$

where $[\mu_k]_{k \in \mathcal{Y}_{0:t}}$ represents concatenated offline and online prototypes. Next, the Top-$K$ highest-entropy instances from each class in task $\mathcal{T}_t$ are identified as hard examples and seamlessly incorporated into PCL training. They dynamically sustain class boundaries and, in conjunction with prototype representations, foster explicit demarcation between both novel and previously learned classes. In fact, the hard instance retrieval strategy may exploit any extant state-of-the-art alternative rather than the specific implementation executed herein, with details provided in Appendix D.2. Eq. 2 is further reformulated as:

$$\mathcal{L}_{PCL}' = \mathbb{E}_{(x_t, y_t) \in \mathcal{T}_t} \left[ -\log \frac{e^{\mathcal{F}_{\theta_t}(x_t^c)^\top \cdot \mu_c / \tau}}{e^{\mathcal{F}_{\theta_t}(x_t^c)^\top \cdot \mu_c / \tau} + \sum_{j \neq c} e^{\mathcal{F}_{\theta_t}(x_t^c)^\top \cdot \mu_j / \tau} + \sum_{c' \neq c} e^{\mathcal{F}_{\theta_t}(x_t^c)^\top \cdot \mathcal{F}_{\theta_t}(x_t^{c'}) / \tau}} \right], \qquad (10)$$

where $c' \in \mathcal{Y}_t$, and $x_t^{c'}$ is drawn from the retrieved hard examples. Moreover, akin to conventional PR, $K$ historical embeddings are stochastically sampled from $\mathcal{N}(\mu_m, \sigma_m^2)$ for each previously observed class $m$, and paired with the current instance $x_t^c$ to constitute negative pairs for PCL. Owing to space limitations, this term is omitted from the denominator in Eq. 10.

### 4.4 FEATURE DRIFT COMPENSATION

While IPAD and PCL exhibit notable efficacy in suppressing feature drift, we further rectify the retained prototypes in the memory buffer $\mathcal{M}$ via post-task drift compensation informed by the current task data $(x_t, y_t) \in \mathcal{T}_t$. The calculation procedure is as follows:

$$\mu'_m = \mu_m + \beta \Delta \mu_m, \ \forall m \in \mathcal{Y}_{0:t-1}, \tag{11}$$

where $\Delta \mu_m = \sum_{(x_t, y_t) \in \mathcal{T}_t} w(x_t, \mu_m)(\mathcal{F}_{\theta_t}(x_t) - \mathcal{F}_{\theta_{t-1}}(x_t))$, and $\beta$ is a hyperparameter controlling compensation intensity. $w(x_t, \mu_m)$ quantifies the proximity between node $x_t$ and prototype $\mu_m$, with closer nodes contributing more to drift compensation. The formal definition is given as follows:

$$w(x_t, \mu_m) = \frac{\mathcal{F}_{\theta_{t-1}}(x_t)^\top \cdot \mu_m}{\sum_{(x'_t, y'_t) \in \mathcal{T}_t} \mathcal{F}_{\theta_{t-1}}(x'_t)^\top \cdot \mu_m}. \tag{12}$$

In this paper, in light of the observation in Figure 1, $\beta$ is assigned an exceedingly small value.

### 4.5 OPTIMIZATION OBJECTIVE

To train the proposed IPAL, the overall optimization objective is as follows:

$$\mathcal{L} = \mathcal{L}'_{PCL} + \gamma \mathcal{L}_{AD}, \tag{13}$$

where $\gamma$ denotes the weighting factor, governing the trade-off between plasticity and stability. Notably, for the base task $\mathcal{T}_0$, the optimization objective is confined to $\mathcal{L}'_{PCL}$, with $\gamma = 0$. We provide a detailed depiction of the training workflow in Appendix F.

## 5 EXPERIMENTS

In this section, we empirically investigate the following questions: **RQ1**) Does IPAL yield performance gains over existing state-of-the-art NECL methods? **RQ2**) Do the proposed components substantively bolster the overall effectiveness of IPAL? **RQ3**) How do hyperparameters such as $\beta$, $|\mathcal{S}_t|$, $K$, and $\gamma$ modulate the performance of IPAL?

### 5.1 EXPERIMENTAL SETUP

**Datasets.** We evaluate IPAL on four node classification benchmark datasets: CS-CL Shchur et al. (2018), CoraFull-CL McCallum et al. (2000), Arxiv-CL Hu et al. (2020) and Reddit-CL Hamilton et al. (2017). Table 1 presents the statistical information of the four benchmark datasets. Following the problem formulation, each dataset is partitioned into a base task $\mathcal{T}_0$ and a series of in-

Table 1: The statistical information of CS-CL, CoraFull-CL, Arxiv-CL, and Reddit-CL.

| Benchmark Datasets ‖ | CS-CL | CoraFull-CL | Arxiv-CL | Reddit-CL |
|---|---|---|---|---|
| # nodes | 18333 | 19793 | 169343 | 232965 |
| # edges | 163788 | 126842 | 1166243 | 114615892 |
| # features | 6805 | 8710 | 128 | 602 |
| # labels | 15 | 70 | 40 | 40 |
| # base classes | 5 | 30 | 20 | 20 |
| # novel classes | 10 | 40 | 20 | 20 |
| # split | 5+5×2 | 30+4×10 | 20+4×5 | 20+4×5 |
| # tasks | 6 | 5 | 5 | 5 |

cremental tasks $\{\mathcal{T}_t\}_{t=1}^{t=N}$ via label-wise stratification. The graph is decomposed into $N + 1$ disjoint subgraphs, each dedicated to a specific training task. All tasks follow a class-wise 6/2/2 split for training/validation/testing. The suffix "-CL" signifies the generated task sequence for CGL. The comprehensive descriptions are delineated in Appendix B.1.

**Implementation Details.** IPAL takes a 2-layer GCN Kipf & Welling (2016) with a hidden dimension of 128, initializing the learning rate with $\eta_0 = 1 \times 10^{-3}$ for the base task and $\eta_{t>0} = 1 \times 10^{-4}$ for incremental tasks. The algorithm implementation and task training are grounded in Continual Graph Learning Benchmark (CGLB) Zhang et al. (2022a), with all experiments executed within the PyTorch 3.10 framework powered by an NVIDIA 3090 GPU. We set the temperature scaling factor $\tau = 0.07$, and the damping factor $\alpha = 0.85$. We perform grid search over $\beta \in [0, 0.5]$, $|\mathcal{S}_t| \in [50, 300]$, and $K \in [5, 30]$. The weighting factor $\gamma$ is tuned from 0.1 to 1.0 in 0.1 increments. Full-graph training is executed on CS-CL and CoraFull-CL, while mini-batch training, with a batch

Table 2: Performance comparison with existing state-of-the-art baselines on CS-CL, CoraFull-CL, Arxiv-CL, and Reddit-CL. The best results are highlighted in bold, and the second-best results are underlined.

| Methods | CS-CL | | CoraFull-CL | | Arxiv-CL | | Reddit-CL | |
|---|---|---|---|---|---|---|---|---|
| | AP/% ↑ | AF/% ↑ | AP/% ↑ | AF/% ↑ | AP/% ↑ | AF/% ↑ | AP/% ↑ | AF/% ↑ |
| Joint | 95.53±0.10 | - | 74.66±0.47 | - | 55.19±0.76 | - | 96.11±0.12 | - |
| Bare | 43.06±3.95 | -63.68±4.84 | 14.63±0.48 | -75.57±0.58 | 16.30±0.18 | -74.41±0.38 | 19.47±0.58 | -96.42±0.69 |
| EWC | 48.11±4.73 | -57.49±5.84 | 15.32±0.95 | -71.17±2.15 | 18.04±1.26 | -67.91±2.50 | 20.77±0.89 | -94.73±1.25 |
| MAS | 53.97±3.67 | -48.25±4.47 | 16.82±0.77 | -65.27±1.19 | 21.66±2.93 | -45.83±6.20 | 20.03±1.46 | -94.13±1.66 |
| LWF | 52.41±3.04 | -52.45±3.67 | 16.51±0.84 | -69.31±1.35 | 16.48±0.33 | -74.73±0.18 | 22.59±2.42 | -92.42±2.94 |
| GEM | 56.56±5.06 | -47.44±6.15 | 19.62±0.23 | -64.62±0.68 | 19.36±0.48 | -68.06±0.59 | 44.99±5.78 | -64.04±7.37 |
| TWP | 55.98±4.17 | -47.98±5.29 | 15.65±0.89 | -73.85±1.82 | 18.58±2.16 | -67.38±4.04 | 23.23±3.21 | -91.65±4.02 |
| ER-GNN | 81.19±1.82 | -17.15±2.20 | 19.54±0.43 | -67.41±0.54 | 27.13±0.34 | -56.55±0.43 | 84.64±2.04 | -14.06±2.53 |
| POLO | 60.68±2.32 | -16.96±3.90 | 23.27±1.60 | **-18.03±1.92** | 32.03±0.75 | -45.48±0.86 | 92.07±1.13 | -1.76±0.79 |
| EFC | 71.65±2.51 | -16.35±3.24 | 38.75±1.85 | -19.55±2.02 | 30.94±0.41 | -40.13±1.48 | 87.49±1.92 | -8.20±2.03 |
| IPAL | **83.07±2.16** | **-12.89±2.50** | **40.69±2.54** | -20.60±0.68 | **33.10±0.84** | -20.59±0.69 | **92.15±0.13** | **-0.27±0.19** |

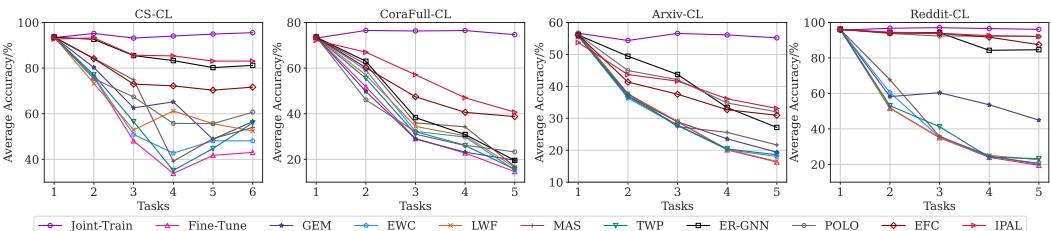

Figure 4: Learning dynamics over the task sequences on CS-CL, CoraFull-CL, Arxiv-CL, and Reddit-CL. The AP is reported on all tasks.

size of 2000, is applied to the larger-scale Arxiv-CL and Reddit-CL datasets. All experiments are run 5 times, with the mean and standard deviation reported.

**Baselines and Evaluation Metrics.** We compare our IPAL with existing state-of-the-art methods, including regularization-based methods (i.e., EWC Kirkpatrick et al. (2017), MAS Aljundi et al. (2018), LWF Li & Hoiem (2017), GEM Lopez-Paz & Ranzato (2017), and TWP Liu et al. (2021)), Non-Exemplar methods (i.e., POLO Wang et al. (2023), and EFC Magistri et al. (2024)), and a classic rehearsal-based method, ER-GNN Zhou & Cao (2021). Furthermore, we consider two canonical baselines: Bare, naive fine-tuning without any auxiliary strategy, and Joint, ideal joint training across tasks, serving as the empirical lower and upper bounds. Average Performance (AP) and Average Forgetting (AF) evaluate the overall classification efficacy and cumulative forgetting across all prior tasks. If a model excels in both metrics, it indicates a balanced trade-off between plasticity and stability. The mathematical definitions are elucidated in Appendix B.2.

## 5.2 RQ1: COMPARISON WITH THE STATE-OF-THE-ART

As shown in Table 2, we compare our IPAL with several state-of-the-art methods on four node classification benchmark datasets. Without intervention for catastrophic forgetting, Bare performs direct fine-tuning on incremental tasks, leading to considerable erosion of prior knowledge. While regularization-based methods mitigate catastrophic forgetting to some extent, their efficacy falls short of rehearsal-based and Non-Exemplar approaches. TWP, tailored for graph-structured tasks, yet in this context, performs comparably to traditional methods. ER-GNN revisits pivotal historical examples to retain previous memory, yet it considers only individual nodes, disregarding the significance of topological structure. In contrast, Non-Exemplar methods replay historical prototypes that encapsulate both class-wise features and topological information, outperforming ER-GNN on CoraFull-CL, Arxiv-CL, and Reddit-CL. However, due to the inherent flaw of conventional PR, feature drift progressively exacerbates with continual model updates. Our proposed IPAL capitalizes on the PCL paradigm to curb feature drift, consistently outstripping existing methods on four benchmark datasets. Notwithstanding a minor decline in AF on CoraFull-CL, IPAL achieves pronounced gains in AP. This discrepancy stems from the inclination of POLO and EFC toward favoring stability over plasticity, whereas our IPAL achieves a more balanced trade-off, yielding superior performance overall. Figure 4 illustrates the learning dynamics on four datasets, highlighting that our IPAL almost

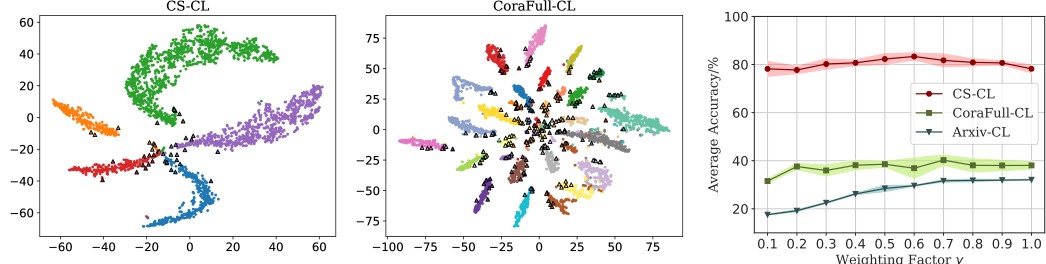

Figure 5: Left two panels: Visualization of class distributions on the base task $\mathcal{T}_0$. Each color corresponds to a specific class, and triangles indicate hard examples. Rightmost panel: Comparison of various $\gamma$ settings on CS-CL, CoraFull-CL, and Arxiv-CL, with AP reported.

invariably outperforms the competing approaches throughout the entire task sequences. Moreover, we assess the computational efficiency of the proposed IPAL in conjunction with the state-of-the-art PR methods, POLO and EFC, with a comprehensive analysis presented in Appendix E.3.

### 5.3 RQ2: Ablation Studies and Visualization

To further validate the proposed components, we conduct ablation studies on CS-CL, CoraFull-CL, and Arxiv-CL. Table 3 reveals the following observations: i) Replacing the topology-aware TIGP with simplistic mean-based prototypes incurs a substantial performance decline, most notably on CS-CL and Arxiv-CL; ii) Eliminating distillation (including IPAD and FD) leads to severe catastrophic forgetting, yielding a performance decline exceeding 8% on each dataset; iii) While FD proves effective in mitigating catastrophic forgetting and feature drift, it may inadvertently impose excessive constraints, hindering the model's capacity to assimilate novel knowledge; iv) DBP takes into account decision-boundary instances in the PCL objective, further promoting inter-class separation.

Moreover, we provide visualizations to elucidate the contributions of the proposed components in a more intuitive manner. Refer to Section Methodology for the analysis of Figure 3. The left two panels in Figure 5 visualizes the class distributions of the base task $\mathcal{T}_0$ on CS-CL and CoraFull-CL, with DBP-identified hard examples located at cluster boundaries, validating our previous analysis. Additional in-

Table 3: Ablation studies on CS-CL, CoraFull-CL, and Arxiv-CL. The AP is reported, with the best results highlighted in bold. FD refers to Feature Distillation.

| TIGP | IPAD | FD | DBP | CS-CL | CoraFull-CL | Arxiv-CL |
|------|------|----|----|-------|-------------|----------|
| ✗ | ✓ | ✗ | ✓ | 79.93±1.85 | 39.98±1.44 | 30.42±0.74 |
| ✓ | ✗ | ✗ | ✓ | 64.07±4.32 | 32.50±2.76 | 16.89±0.25 |
| ✓ | ✗ | ✓ | ✓ | 72.79±2.70 | 36.04±2.83 | 27.25±0.71 |
| ✓ | ✓ | ✗ | ✗ | 75.94±3.05 | 33.63±2.58 | 31.06±0.42 |
| ✓ | ✓ | ✗ | ✓ | **83.07±2.16** | **40.69±2.54** | **33.10±0.84** |

sights into the underpinning rationale of the proposed components' architecture are elaborated in Appendix D. Concurrently, we furnish a comprehensive efficiency evaluation and scalability analysis in Appendix E.

### 5.4 RQ3: Parameter Analysis

A grid search over $[0.1, 1.0]$ is carried out on the validation sets of CS-CL, CoraFull-CL, and Arxiv-CL to examine performance sensitivity to the weighting factor $\gamma$. As shown in the rightmost panel in Figure 5, the optimal performance is attained with a weighting factor of 0.6 or 0.7 on small-scale datasets, whereas larger datasets require a higher weight. Overall, smaller $\gamma$ hinders task retention (i.e., stability ↓, plasticity ↑), while larger values impose undue constraints (i.e., stability ↑, plasticity ↓). Empirically, a value around 0.7 strikes a favorable trade-off between plasticity and stability, consistently yielding superior performance on all datasets and streamlining the hyperparameter tuning process. Appendix C provides further analyses of the residual parameters.

## 6 CONCLUSIONS AND FUTURE WORKS

In this paper, we empirically revealed that PCL substantially attenuates feature drift compared to conventional PR by harnessing the intrinsic relational topology between instances and prototypes. We further proposed IPAL, a novel NECGL paradigm built upon PCL. To be specific, we evaluated node importance via the PageRank algorithm and generated the topology-aware TIGP to promote PCL training. To address the inherent feature drift and catastrophic forgetting in NECGL, IPAD was proposed to regularize the relational structure between instances and prototypes, providing greater flexibility while seamlessly aligning with the PCL objective function. Furthermore, the DBP mechanism was leveraged to mine hard examples, mitigating inter-class ambiguity and fostering more pronounced inter-class separability. Extensive empirical evaluations on four node classification benchmark datasets, including comparative, ablation, and parameter studies, demonstrated that IPAL consistently outperformed existing state-of-the-art methods. Our future work will seek to adapt IPAL to online settings with streaming data, promoting its scalability and real-world viability.

## REPRODUCIBILITY STATEMENT

All resources necessary to reproduce the results are provided in the main text, appendix, and supplementary materials. In particular, the full implementation of our models and algorithms is available as anonymous downloadable supplementary material; all assumptions and formal proofs underpinning theoretical claims are presented in Section 1 and Appendix A; all datasets used in the experiments, together with detailed preprocessing procedures and data splits, are comprehensively documented in Section 5.1 and Appendix B.1.

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

# APPENDIX

## A  THEORETICAL ANALYSIS ON FEATURE DRIFT

**Notation.** Let $\mathcal{F}_\theta(\cdot) : x \mapsto f$ denote the GNN encoder, inherently Lipschitz continuous with respect to the input Ruiz et al. (2021). Access pertaining to task $\mathcal{T}_t$ is confined to the current task data $(x_t, y_t) \in \mathcal{T}_t$ and the prototypes $(\mu_m, \Sigma_m)$ stored in the memory buffer $\mathcal{M}$, where $m \in \mathcal{Y}_{0:t-1}$. After training on task $\mathcal{T}_t$, the GNN encoder is updated from $\theta_{t-1}$ to $\theta_t$. Consider the samples $(x_m, y_m) \in \mathcal{T}_{0:t-1}$ from previous tasks. Let the feature distributions follow a Gaussian distribution, where $\mathcal{F}_{\theta_t}(x_m) \sim \mathcal{P}_t(f) = \mathcal{N}(\mu'_m, \Sigma'_m)$ and $\mathcal{F}_{\theta_{t-1}}(x_m) \sim \mathcal{P}_{t-1}(f) = \mathcal{N}(\mu_m, \Sigma_m)$. Note that the notation here differs slightly from that in the main text, such as the covariance $\Sigma_m$ and the diagonal covariance $\sigma_m^2$.

In Non-Exemplar Continual Learning (NECL), Prototype Contrastive Learning (PCL) induces less drift than conventional Prototype Replay (PR). To validate this theoretically, we adopt the Kullback-Leibler (KL) divergence to quantify feature drift and investigate the evolution of feature distributions from prior tasks during the assimilation of new tasks for both PCL and PR, as follows:

$$D_{KL}(\mathcal{P}_{t-1} \| \mathcal{P}_t) = \int \mathcal{P}_{t-1}(f) \log \frac{\mathcal{P}_{t-1}(f)}{\mathcal{P}_t(f)} \, df. \tag{14}$$

The goal is to prove that the KL divergence for PCL is theoretically less than that for PR, as formulated below:

**Theorem 1** *In NECL, assuming Gaussian feature distributions with $\mathcal{F}_{\theta_t}(x_m) \sim \mathcal{P}_t(f) = \mathcal{N}(\mu'_m, \Sigma'_m)$ and $\mathcal{F}_{\theta_{t-1}}(x_m) \sim \mathcal{P}_{t-1}(f) = \mathcal{N}(\mu_m, \Sigma_m)$, where $\Sigma'_m$ and $\Sigma_m$ are positive definite, PCL incurs a smaller feature drift than PR:*

$$D_{KL}(\mathcal{P}_{t-1} \| \mathcal{P}_t)_{PCL} < D_{KL}(\mathcal{P}_{t-1} \| \mathcal{P}_t)_{PR}. \tag{15}$$

**Proof.** For the $n$-dimensional multivariate Gaussian distribution, the expression has a closed-form solution as follows:

$$D_{KL}(\mathcal{P}_{t-1} \| \mathcal{P}_t) = \frac{1}{2} \left( \log \frac{\det \Sigma'_m}{\det \Sigma_m} + \text{tr}(\Sigma_m'^{-1} \Sigma_m) + (\mu'_m - \mu_m)^\top \Sigma_m'^{-1} (\mu'_m - \mu_m) - n \right). \tag{16}$$

To prove Theorem 1, it suffices to show that for prior feature distributions, PCL yields smaller $\|\mu'_m - \mu_m\|_2^2$ and $\text{tr}(\Sigma_m'^{-1} \Sigma_m - \text{I})$ than PR.

PCL updates $\theta_t$ on task $\mathcal{T}_t$ by optimizing $\mathcal{L}_{PCL}$, with the feature gradient computed as follows:

$$\nabla_{f_{\theta_t}} \mathcal{L}_{PCL} \propto -\frac{1}{\tau} \left[ \mu_c - \sum_{k \in \mathcal{Y}_{0:t}} \frac{e^{f_{\theta_t}^\top \cdot \mu_k / \tau}}{\sum_{k' \in \mathcal{Y}_{0:t}} e^{f_{\theta_t}^\top \cdot \mu_{k'} / \tau}} \mu_k \right]. \tag{17}$$

The gradient for the model parameters $\theta_t$ is computed as follows:

$$\nabla_\theta \mathcal{L}_{PCL} = \nabla_{f_{\theta_t}} \mathcal{L}_{PCL} \cdot \nabla_\theta \mathcal{F}_{\theta_t}(x_t). \tag{18}$$

Considering $(x_m, y_m) \in \mathcal{T}_{0:t-1}$, we have:

$$\mathcal{F}_{\theta_t}(x_m) = \mathcal{F}_{\theta_{t-1} + \Delta\theta}(x_m). \tag{19}$$

Here, considering an infinitesimal update step $\Delta\theta$, it can be approximated by the first-order Taylor expansion Zenke et al. (2017) as follows:

$$\mathcal{F}_{\theta_t}(x_m) \approx \mathcal{F}_{\theta_{t-1}}(x_m) + \nabla_\theta \mathcal{F}_{\theta_{t-1}}(x_m) \cdot \Delta\theta. \tag{20}$$

Furthermore, we obtain:

$$\mu'_m \approx \mu_m + \mathbb{E}_{(x_m, y_m) \in \mathcal{T}_{0:t-1}} \left[ \nabla_\theta \mathcal{F}_{\theta_{t-1}}(x_m) \cdot \Delta\theta \right]. \tag{21}$$

$\mathcal{L}_{PCL}$ regularizes the negative sample gradients in Eq. 17 by minimizing $\mathcal{F}_{\theta_t}(x_t)^\top \cdot \mu_m$, encouraging $\Delta\theta$ to drive $\mathcal{F}_{\theta_t}(x_m)$ closer to $\mathcal{F}_{\theta_{t-1}}(x_m)$. Thus, $\mathcal{L}_{PCL}$ essentially regularizes $\|\mu'_m - \mu_m\|_2^2$, effectively alleviating feature shift.

PR optimizes $\mathcal{L}_{PR}$ during training, with the feature gradient computed as follows:

$$\nabla_{f_{\theta_t}} \mathcal{L}_{PR} \propto \sum_{k \in \mathcal{Y}_{0:t}} \left( \frac{e^{f_{\theta_t}^\top \cdot w_k}}{\sum_{k' \in \mathcal{Y}_{0:t}} e^{f_{\theta_t}^\top \cdot w_{k'}}} - \mathbb{1}\{k = y_t^c\} \right) w_k, \tag{22}$$

where $w_k$ denotes the class-$k$ weight of the linear classifier $g_{\phi_t}(\cdot)$. Since PR exclusively emphasizes the classification accuracy of the new sample $(x_t, y_t) \in \mathcal{T}_t$ without explicitly regularizing $\Delta\theta$ to maintain the alignment between $\mathcal{F}_{\theta_t}(x_m)$ and $\mathcal{F}_{\theta_{t-1}}(x_m)$, it may exacerbate the deviation, leading to an increase in $\|\mu'_m - \mu_m\|_2^2$.

On the other hand, the covariance of past tasks is defined as:

$$\Sigma'_m = \mathbb{E}_{(x_m, y_m) \in \mathcal{T}_{0:t-1}} \left[ \left( \mathcal{F}_{\theta_t}(x_m) - \mu'_m \right) \left( \mathcal{F}_{\theta_t}(x_m) - \mu'_m \right)^\top \right]. \tag{23}$$

According to our preceding analysis, PCL encourages alignment between $\mathcal{F}_{\theta_t}(x_m)$ and $\mathcal{F}_{\theta_{t-1}}(x_m)$, whereas PR, lacking such a constraint, allows $\Sigma_m$ to expand uncontrollably.

Therefore, we conclude that PCL yields smaller $\|\mu'_m - \mu_m\|_2^2$ and $\text{tr}(\Sigma_m'^{-1}\Sigma_m - I)$ than PR, thereby substantiating the validity of Theorem 1.

# B    EXTENDED DETAILS ON EXPERIMENTAL CONFIGURATION

## B.1    ADDITIONAL DESCRIPTIONS ON THE DATASETS

Four node classification benchmark datasets are engaged in this paper, with the following detailed descriptions:

**CS-CL (Shchur et al., 2018).** Coauthor CS is a co-authorship graph based on the Microsoft Academic Graph from the KDD Cup 2016 challenge. Nodes stand for authors, linked by edges in cases of co-authorship. Node attributes encode paper keywords, and class labels signify the author's principal research domains. In this study, we divide the dataset into a base task with 5 classes and 5 incremental tasks, each containing 2 of the remaining 10 classes.

**CoraFull-CL (McCallum et al., 2000).** CoraFull is a more complete citation network dataset than the commonly used 7-class subset, with nodes as papers, labels as topics, and edges as citation links. All papers are classified into 70 discrete topics, from which 30 are designated to constitute the base task, while the remaining 40 are evenly partitioned into 4 incremental tasks of 10 topics each.

**Arxiv-CL (Hu et al., 2020).** OGB-Arxiv is a paper citation network of Arxiv papers extracted from the Microsoft Academic Graph. Each node is an Arxiv paper, with directed edges indicating citations between papers. The skip-gram model Mikolov et al. (2013) is applied to extract word embeddings from titles and abstracts, which are then employed to define node attributes. The dataset comprises 40 subject areas from Arxiv Computer Science papers. The first 20 areas form the base task, and the remaining 20 areas are grouped into 4 incremental tasks of 5 areas each.

**Reddit-CL (Hamilton et al., 2017).** Reddit comprises posts made in September 2014, with each node labeled by its associated subreddit. 41 large communities are taken into account to construct a post-to-post graph, with edges defined by user comments on both posts. Node attributes include the post title, average comment embedding, post score, and comment count. Following Zhang et al. (2022a), we exclude the $41st$ class, then treat the first 20 as the base task and group the remaining 20 into 4 incremental tasks of 5 classes each.

## B.2    MATHEMATICAL DEFINITIONS FOR THE EVALUATION METRICS

Average Performance (AP) and Average Forgetting (AF) quantify the overall classification performance and the extent of catastrophic forgetting, respectively. Their mathematical definitions are as follows:

$$\text{AP}_t = \frac{\sum_{i=1}^{t} \mathbf{M}_{t,i}}{t}, \quad \text{AF}_t = \frac{\sum_{i=1}^{t-1} \mathbf{M}_{t,i} - \mathbf{M}_{i,i}}{t - 1}, \tag{24}$$

where $\mathbf{M}$ denotes the lower triangular performance matrix (refer to Figure 3 in the main text). $\mathbf{M}_{t,i}$ is the prediction accuracy on task $\mathcal{T}_i$ after training on task $\mathcal{T}_t$, with $t$ indexed from 1 for convenience.

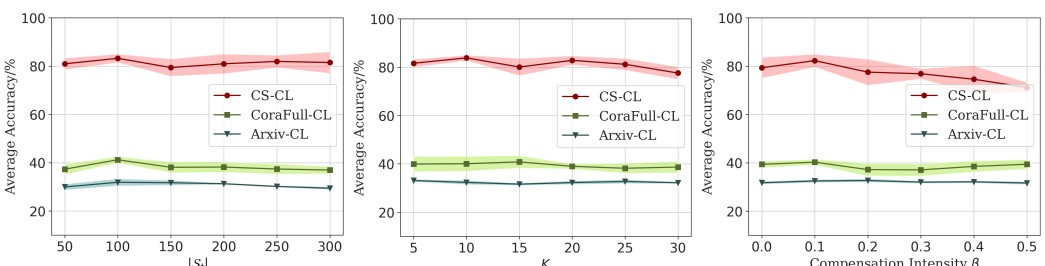

Figure 6: Parameter searches for $|\mathcal{S}_t|$, $K$, and $\beta$ on CS-CL, CoraFull-CL, and Arxiv-CL, with AP reported.

Table 4: Comparison of PageRank- and degree-centrality weighting on CS-CL and Reddit-CL, with the best results highlighted in bold.

| Methods | CS-CL | | Reddit-CL | |
|---|---|---|---|---|
| | AP/% ↑ | AF/% ↑ | AP/% ↑ | AF/% ↑ |
| Degree Centrality | 80.16±2.00 | -15.95±2.10 | 91.60±0.35 | -0.77±0.38 |
| PageRank | **83.07±2.16** | **-12.89±2.50** | **92.15±0.13** | **-0.27±0.19** |

Owing to the inherent trade-off between AP and AF, an elevated AP may compromise AF, and vice versa. Both metrics should be taken into account in performance evaluation.

## C    ADDITIONAL ANALYSIS FOR THE RESIDUAL PARAMETERS

Figure 6 presents a grid search over hyperparameters $|\mathcal{S}_t|$, $K$, and Compensation Intensity $\beta$, illustrating the performance variation of IPAL under different settings. The empirical evidence supports the following conclusions: i) A larger subset sampling size $|\mathcal{S}_t|$ does not necessarily lead to better performance, since increasing $|\mathcal{S}_t|$ can introduce stricter distillation constraints that may adversely affect the model's plasticity. In this study, $|\mathcal{S}_t|$ is fixed at 100 to achieve a trade-off between stability and plasticity; ii) Increasing the number of hard examples $K$ sharpens novel class boundaries but weakens class prototype influence. For class balance, $K$ historical embeddings per old class are drawn from $\mathcal{N}(\mu_m, \sigma_m^2)$, with larger $K$ potentially introducing noise. Given IPAL's insensitivity to $K$ on large-scale datasets, we fix $K = 10$; iii) Feature drift challenges prototype-based NECL methods, but thanks to PCL's robustness, IPAL achieves competitive performance even without drift compensation (e.g., $\beta = 0$). Setting $\beta = 0.1$ yields improved results across all three datasets. With further increases in $\beta$, performance deteriorates, empirically supporting Theorem 1 that excessive compensation is unnecessary for PCL.

## D    ADDITIONAL ANALYSIS ON COMPONENT ARCHITECTURE

### D.1    PAGERANK VS. DEGREE CENTRALITY

To elucidate the rationale behind PageRank, we compare PageRank-based weighting with degree-centrality weighting on the small-scale CS-CL and large-scale Reddit-CL benchmarks. The empirical evidence in Table 4 demonstrates that the PageRank weighting scheme applied in TIGP consistently exceeds its degree-centrality counterpart. Degree centrality, inherently limited to local connectivity, contrasts with PageRank, which harnesses stochastic diffusion to reveal more comprehensive and globally coherent structural patterns. Moreover, in task graphs, degree centrality assigns null influence to potentially isolated nodes, whereas PageRank, via its damping factor $\alpha$, systematically propagates influence in a principled and globally coherent manner.

Table 5: Performance comparison of various hard instance retrieval strategies on CS-CL and Reddit-CL. The best results are highlighted in bold, and the second-best results are underlined.

| Methods | CS-CL | | Reddit-CL | |
|---|---|---|---|---|
| | AP/% ↑ | AF/% ↑ | AP/% ↑ | AF/% ↑ |
| w/o DBP | 75.94±3.05 | -21.05±4.00 | 88.08±1.01 | -4.45±1.65 |
| w/ Margin | **86.20±2.34** | **-8.60±2.25** | **93.38±0.28** | -0.40±0.14 |
| w/ Energy | 79.81±2.14 | -16.54±1.69 | 90.92±1.03 | -0.80±0.32 |
| w/ Entropy | 83.07±2.16 | -12.89±2.50 | 92.15±0.13 | **-0.27±0.19** |

## D.2 HARD INSTANCE RETRIEVAL STRATEGY

The hard instance retrieval strategy in the proposed DBP mechanism is not restricted to the entropy-based approach and can be seamlessly supplanted with other state-of-the-art alternatives, such as margin- and energy-based methods. We examine the effect of various hard instance retrieval strategies on the performance of the proposed IPAL framework across the CS-CL and Reddit-CL benchmark datasets. Here, for a node in task $\mathcal{T}_t$ with label $c$ and embedding $f_{\theta_t}^c$, we define the margin- and energy-based strategies as follows.

☐ **Margin-based method.** We adopt EL2N (Sabbineni et al., 2023) as a margin-oriented metric, providing a reliable measure of example hardness via gradient norm estimation. Formally, EL2N is defined as follows.

$$\text{EL2N}(x_t^c) = \|softmax(f_{\theta_t}^{c \, \mathrm{T}} \cdot [\mu_k]_{k \in \mathcal{Y}_{0:t}}) - \text{onehot}(c)\|_2, \tag{25}$$

where $[\mu_k]_{k \in \mathcal{Y}_{0:t}}$ denotes the concatenation of offline and online prototypes, and $\text{onehot}(c)$ represents the one-hot encoding of label $c$.

☐ **Energy-based method.** The free energy (Zhang et al., 2024) quantifies the likelihood of an input sample, with higher energy values indicating less favorable states for the model and, consequently, greater learning difficulty. The free energy is computed as follows.

$$F(x_t^c) = -\log \sum_{k \in \mathcal{Y}_{0:t}} e^{-E(x_t^c, k)}, \tag{26}$$

where $E(x_t^c, k) = -f_{\theta_t}^{c \, \mathrm{T}} \cdot [\mu_k]_{k \in \mathcal{Y}_{0:t}}$.

Table 5 illustrates that hard instances retrieved via distinct metrics exhibit variation, thereby shaping PCL's learning dynamics. The margin-based strategy yields the greatest gains by exploiting explicit label information, whereas free-energy- and entropy-based strategies also enhance performance. Ablating the DBP mechanism substantially impairs performance, highlighting that hard instances, serving as surrogates for the decision boundary, improve inter-class separability and strengthen CGL.

## E EFFICIENCY AND SCALABILITY ANALYSIS

In this section, we examine the efficiency and scalability of the proposed components on the small-scale CS-CL and large-scale Reddit-CL benchmarks.

### E.1 COMPUTATIONAL EFFICIENCY OF PAGERANK

First, we analyze the time complexity of the PageRank algorithm. For task $\mathcal{T}_t$, the computational cost is primarily dominated by the matrix multiplication between the transition matrix $\mathbf{P}_t = \mathbf{A}_t^{\mathrm{T}} \mathbf{D}^{-1} \in \mathbb{R}^{|\mathcal{V}_t| \times |\mathcal{V}_t|}$ and the PageRank vector $\mathbf{R}_t \in \mathbb{R}^{|\mathcal{V}_t|}$. When $\mathbf{P}_t$ is treated as a dense matrix, $n$ iterations incur a time complexity of $\mathcal{O}(n|\mathcal{V}_t|^2)$, whereas exploiting the sparsity of $\mathbf{P}_t$ reduces the time complexity to $\mathcal{O}(n|\mathcal{E}_t|)$.

On this basis, we profile the runtime of PageRank under dense matrix computations. For CS-CL, PageRank runs on a single NVIDIA GeForce RTX 3090 GPU, whereas for Reddit-CL, due to out-of-memory (OOM) constraints, computations run on a CPU (Intel Xeon Gold 6226R, 64 cores, 2.90

Table 6: Task graph statistics and PageRank runtime on CS-CL and Reddit-CL.

| Benchmark Datasets | Task Statistics & Runtime | $\mathcal{T}_0$ | $\mathcal{T}_1$ | $\mathcal{T}_2$ | $\mathcal{T}_3$ | $\mathcal{T}_4$ | $\mathcal{T}_5$ |
|---|---|---|---|---|---|---|---|
| CS-CL | # nodes | 5043 | 2564 | 1699 | 1562 | 2453 | 5012 |
| | # edges | 39884 | 16602 | 14000 | 13686 | 16576 | 39148 |
| | Runtime/ms | 8.86 | 5.09 | 4.83 | 4.29 | 5.08 | 8.29 |
| Reddit-CL | # nodes | 133970 | 26358 | 21634 | 17380 | 28511 | - |
| | # edges | 56195430 | 5403916 | 4637618 | 10989194 | 15995550 | - |
| | Runtime/s | 73.31 | 4.06 | 2.41 | 1.40 | 4.22 | - |

Table 7: Scalability evaluation of $|\mathcal{S}_t|$ and $K$ on CS-CL and Reddit-CL, with average epoch runtime and peak memory footprint on the incremental tasks $\mathcal{T}_{t>0}$ reported.

| Benchmark Datasets | Metrics | IPAD-$|\mathcal{S}_t|$ | | | DBP-$K$ | | |
|---|---|---|---|---|---|---|---|
| | | 100 | 200 | 300 | 10 | 20 | 30 |
| CS-CL | Runtime/ms | 3.52 | 3.79 | 4.19 | 2.30 | 2.37 | 1.93 |
| | Peak Memory/MB | 3.20 | 6.52 | 9.83 | 10.75 | 18.19 | 26.27 |
| Reddit-CL | Runtime/s | 0.08 | 0.09 | 0.09 | 0.04 | 0.04 | 0.05 |
| | Peak Memory/MB | 73.29 | 143.78 | 214.69 | 205.86 | 387.09 | 561.10 |

GHz) with 251 GiB of RAM. As shown in Table 6, the computational cost of PageRank on the small-scale benchmark CS-CL is on the order of microseconds and negligible during training. On the large-scale Reddit-CL benchmark, PageRank computation for the base task $\mathcal{T}_0$, which involves nodes from 20 classes and their interconnecting edges, incurs a slightly higher cost but remains only a few seconds for incremental tasks $\mathcal{T}_{t>0}$. Crucially, PageRank can be precomputed prior to the training procedure of each task, allowing the use of the resulting $\mathbf{R}_t$ during training without any additional overhead.

### E.2 SCALABILITY ANALYSIS OF $|\mathcal{S}_t|$ AND $K$

Furthermore, we investigate the scalability of the IPAD and DBP components by systematically varying the hyperparameters $|\mathcal{S}_t|$ and $K$, respectively, and recording their average epoch runtime and peak memory footprint on the incremental tasks $\mathcal{T}_{t>0}$. Table 7 indicates that increasing $|\mathcal{S}_t|$ or $K$ incurs only a marginal effect on runtime. Although peak memory consumption experiences a modest escalation with larger hyperparameter values, the associated overhead remains inconsequential relative to the 24 GB capacity of an NVIDIA GeForce RTX 3090 GPU. Hence, the proposed components incur fully tractable computational overhead, thereby underscoring their inherent scalability.

### E.3 COMPUTATIONAL EFFICIENCY COMPARISON WITH STATE-OF-THE-ART PROTOTYPE REPLAY METHODS

Table 8 provides a comparison of the proposed IPAL against the state-of-the-art Prototype Replay methods, POLO (Wang et al., 2023) and EFC (Magistri et al., 2024), in terms of average epoch-wise training time and peak memory footprint on CS-CL and Reddit-CL. Empirical results demonstrate that IPAL achieves efficiency on par with or exceeding that of EFC in both training time and memory footprint, while substantially outperforming POLO. Drawing on the evidence from Table 2, IPAL is found to deliver superior CGL performance while incurring a comparatively lower computational overhead.

## F PSEUDOCODE FOR TRAINING IPAL

Algorithm 1 provides a detailed depiction of the training workflow.

Table 8: Task-level average epoch runtime and peak memory footprint on CS-CL and Reddit-CL, in comparison with state-of-the-art Prototype Replay methods.

| Benchmark Datasets | Methods | $\mathcal{T}_0$ | | $\mathcal{T}_1$ | | $\mathcal{T}_2$ | |
|---|---|---|---|---|---|---|---|
| | | Runtime/s | Peak Memory/MB | Runtime/s | Peak Memory/MB | Runtime/s | Peak Memory/MB |
| CS-CL | POLO | 0.01 | 300.10 | 0.07 | 158.99 | 0.10 | 124.31 |
| | EFC | 0.01 | 301.88 | 0.02 | 160.56 | 0.02 | 125.56 |
| | IPAL | 0.02 | 300.81 | 0.02 | 160.21 | 0.02 | 125.23 |
| Reddit-CL | POLO | 5.31 | 1483.81 | 1.86 | 616.96 | 1.85 | 588.23 |
| | EFC | 5.08 | 1463.27 | 0.62 | 721.59 | 0.53 | 686.50 |
| | IPAL | 5.86 | 1059.96 | 0.57 | 425.54 | 0.48 | 403.87 |

| Benchmark Datasets | Methods | $\mathcal{T}_3$ | | $\mathcal{T}_4$ | | $\mathcal{T}_5$ | |
|---|---|---|---|---|---|---|---|
| | | Runtime/s | Peak Memory/MB | Runtime/s | Peak Memory/MB | Runtime/s | Peak Memory/MB |
| CS-CL | POLO | 0.11 | 119.13 | 0.12 | 155.69 | 0.14 | 305.49 |
| | EFC | 0.02 | 120.44 | 0.02 | 158.90 | 0.03 | 307.89 |
| | IPAL | 0.02 | 119.74 | 0.03 | 156.61 | 0.03 | 306.62 |
| Reddit-CL | POLO | 1.74 | 499.49 | 3.01 | 698.72 | - | - |
| | EFC | 0.45 | 577.37 | 0.79 | 820.20 | - | - |
| | IPAL | 0.39 | 340.07 | 0.72 | 482.59 | - | - |

---

**Algorithm 1** Training procedure for our IPAL.

---

**Input**: Task sequence $\mathcal{T} = \{\mathcal{T}_0, \mathcal{T}_1, ..., \mathcal{T}_N\}$, GNN encoder $\mathcal{F}_\theta(\cdot)$, memory buffer $\mathcal{M}$, weighting factor $\alpha$, base task learning rate $\eta_0$, incremental task learning rate $\eta_{t>0}$, number of epochs $E$.
**Output**: Predicted labels for test nodes from all previously learned tasks.

1: **for** $t = 0, 1, ..., N$ **do**
2:    **for** $epoch = 1, 2, ..., E$ **do**
3:      **if** $t = 0$ **then**
       // Initial training on $\mathcal{T}_0$.
4:        Compute $\mathcal{L}'_{PCL}$ according to Eq. 10 in the main text.
5:        $\theta_t \leftarrow \theta_t - \eta_0 \nabla_{\theta_t} \mathcal{L}'_{PCL}$.
6:      **else**
       // Incremental training on $\mathcal{T}_{t>0}$.
7:        Compute $\mathcal{L}$ according to Eq. 13 in the main text.
8:        $\theta_t \leftarrow \theta_t - \eta_t \nabla_{\theta_t} \mathcal{L}$.
9:      **end if**
10:   **end for**
11:   Perform drift compensation on prior class prototypes $\{\mu_m\}_{m \in \mathcal{Y}_{0:t-1}}$ according to Eq. 11 in the main text.
12:   Generate TIGP $\{\mathcal{N}(\mu_c, \sigma_c^2)\}_{c \in \mathcal{Y}_t}$ according to Eq. 4 in the main text, and allocate them to the memory buffer $\mathcal{M}$.
     // Testing phase.
13:   Predict node labels for all prior task graphs:
     $\hat{y}(x) = \arg\max_{k \in \mathcal{Y}_{0:t}} \mathcal{F}_{\theta_t}(x)^\top \cdot [\mu_k], \forall x \in \mathcal{T}_{0:t}$.
14: **end for**
15: **return** $\hat{y}(x)$.

---

# G LIMITATIONS

While our method exhibits competitive performance in Non-Exemplar Continual Graph Learning, several limitations merit further exploration: i) Existing approaches, including ours, assume a unified dataset where tasks are simulated by class-based subgraph partitioning within a single graph. However, in real-world scenarios involving heterogeneous domains, the cross-domain generalizability remains to be validated; ii) Our method is trained offline, but its applicability to online settings, where data arrives in a mini-batch streaming manner, remains to be further investigated.

## H    THE USE OF LARGE LANGUAGE MODELS (LLMS)

The authors declare that LLMs (e.g., GPT-5 and Grok 4) were employed exclusively for linguistic polishing during manuscript preparation, without any involvement in research ideation, methodology, experimentation, or other aspects.

