# OpenReview forum: "Instance-Prototype Affinity Learning for Non-Exemplar Continual Graph Learning"
_ICLR.cc/2026/Conference — Submitted to ICLR 2026_

### Official Review · Reviewer_KRRQ · 2025-10-22

**Soundness:** 3
**Presentation:** 2
**Contribution:** 2
**Rating:** 4
**Confidence:** 3

**Summary:**

The paper studies continual graph learning without storing past samples and focuses on reducing catastrophic forgetting caused by prototype drift. It identifies three pitfalls: simple mean prototypes that ignore node importance, feature distillation that overconstrains the space, and single prototypes that blur class boundaries. The method, IPAL, combines Topology Integrated Gaussian Prototypes that weight nodes by PageRank, Instance Prototype Affinity Distillation that aligns instance to prototype relations, and a Decision Boundary Perception mechanism that stresses high uncertainty cases inside a prototype contrastive learning objective. On four node classification benchmarks, the approach improves average accuracy and lowers forgetting compared with regularization, rehearsal, and prior non exemplar methods, approaching or surpassing replay while maintaining a better balance between stability and plasticity.

**Strengths:**

S1: Well-grounded motivation: Replaces feature-level alignment with instance–prototype relations and employs PCL to suppress drift; the rationale is clear and verifiable.

S2: Cohesive design: TIGP, IPAD, and DBP respectively address topology-aware weighting, the stability–plasticity trade-off, and boundary shaping; experiments show consistent gains over diverse baselines.

**Weaknesses:**

W1. Cost and scalability: PageRank seems manageable, but on very large graphs with frequent tasks the online prototype updates and DBP mining may bottleneck throughput. Please add end-to-end runtime, peak memory, scaling curves on fixed hardware, and sensitivity/throughput analyses for $S_t$, $K$, and $\beta$.

W2. Theoretical support: The KL analysis is intuitive but lacks assumptions that yield tighter PCL drift bounds and provable adaptivity of $\gamma$ and $\beta$.

W3. Boundary hard-example selection: DBP uses entropy as a boundary proxy, yet class-wise miscalibration can bias a single threshold. Provide class-level calibration/uncertainty statistics and compare mining by entropy, margin, energy, and temperature-calibrated scores to establish effectiveness.

**Questions:**

see the weaknesses above

---

> ### Author Response · Authors · 2025-11-20
> **Responses to W1 (Part 1)**
>
> We thank the reviewer for the insightful comments.
>
> Following your suggestion, we evaluated the efficiency of the framework and its constituent components, including PageRank, TIGP, IPAD, and DBP.
>
>
> **PageRank:** First, we investigated the time complexity of the PageRank algorithm. For task $\mathcal{T} _ t$, the computational cost is primarily dominated by the matrix multiplication between the transition matrix $\mathbf{P} _ t=\mathbf{A} _ t^{\mathrm{T}}\mathbf{D}^{-1}\in\mathbb{R}^{|\mathcal{V} _ t|\times|\mathcal{V} _ t|}$ and the PageRank vector $\mathbf{R} _ t\in\mathbb{R}^{|\mathcal{V} _ t|}$. When $\mathbf{P} _ t$ is treated as a dense matrix, $n$ iterations incur a time complexity of $\mathcal{O}(n|\mathcal{V} _ t|^2)$; leveraging the sparsity of $\mathbf{P} _ t$ reduces the time complexity to $\mathcal{O}(n|\mathcal{E} _ t|)$. Furthermore, we evaluated the computational efficiency of PageRank on both the small-scale CS-CL and the large-scale Reddit-CL benchmark datasets, considering dense matrix computations. PageRank on CS-CL was executed on a single NVIDIA GeForce RTX 3090 GPU, whereas due to out-of-memory (OOM) issues with Reddit-CL, computations were performed on a CPU (Intel Xeon Gold 6226R, 64 cores, 2.90 GHz) with 251 GiB of RAM. Experimental results are shown below. On the small-scale benchmark dataset CS-CL, the computational cost of PageRank is on the order of microseconds and is negligible during training. For the large-scale benchmark Reddit-CL, as the base task $\mathcal{T} _ 0$ involves nodes from 20 classes and their interconnecting edges, the PageRank computation incurs a slightly higher cost; yet for incremental tasks $\mathcal{T}_{t>0}$, it remains only a few seconds. Crucially, PageRank can be precomputed prior to each task, allowing the use of the resulting $\mathbf{R}_t$ during training without any additional overhead.
>
> | Benchmark Datasets | Info.      | $\mathcal{T}_0$ | $\mathcal{T}_1$ | $\mathcal{T}_2$ | $\mathcal{T}_3$ | $\mathcal{T}_4$ | $\mathcal{T}_5$ |
> |--------------------|------------|-----------------|-----------------|-----------------|-----------------|-----------------|-----------------|
> | CS-CL              | # nodes    | 5043            | 2564            | 1699            | 1562            | 2453            | 5012            |
> |                    | # edges    | 39884           | 16602           | 14000           | 13686           | 16576           | 39148           |
> |                    | Runtime/ms | 8.86            | 5.09            | 4.83            | 4.29            | 5.08            | 8.29            |
> | Reddit-CL          | # nodes    | 133970          | 26358           | 21634           | 17380           | 28511           | -               |
> |                    | # edges    | 56195430        | 5403916         | 4637618         | 10989194        | 15995550        | -               |
> |                    | Runtime/s  | 73.31           | 4.06            | 2.41            | 1.40            | 4.22            | -               |

---

> ### Author Response · Authors · 2025-11-20
> **Responses to W1 (Part 2)**
>
> We next report the average per-epoch runtime and peak memory usage of the proposed TIGP, IPAD, and DBP components on CS-CL and Reddit-CL. Notably, TIGP here involves only the computation of online prototypes, while DBP includes both hard example retrieval and PCL loss computation.
>
> All experiments were conducted on a single NVIDIA GeForce RTX 3090 GPU. The three tables below report the task-level average per-epoch runtime and peak memory usage. On the small-scale CS-CL benchmark, each component exhibits microsecond-level runtimes and low memory consumption. On the large-scale Reddit-CL benchmark, the base task $\mathcal{T} _ 0$, which contains an order of magnitude more nodes and edges than incremental tasks $\mathcal{T} _ {t>0}$, incurs higher computational and memory costs, while the overhead for incremental tasks remains substantially lower. Notably, all reported runtimes and memory usage are fully manageable on a single 24GB RTX 3090 GPU.
>
> - Average per-epoch runtime and peak memory usage of TIGP:
>
> | Benchmark Datasets | Metrics     | $\mathcal{T}_0$ | $\mathcal{T}_{1}$ | $\mathcal{T}_{2}$ | $\mathcal{T}_{3}$ | $\mathcal{T}_{4}$ | $\mathcal{T}_{5}$ |
> |--------------------|----------------|-----------------|-------------------|-------------------|-------------------|-------------------|-------------------|
> | CS-CL              | Runtime/ms     | 1.10             | 0.73              | 0.53              | 0.55              | 0.55              | 0.53              |
> |                    | Peak Memory/MB | 2.47            | 2.99              | 0.54              | 0.85              | 3.10              | 5.57              |
> | Reddit-CL          | Runtime/s      | 0.24            | 0.01              | 0.01              | 0.01              | 0.01              | -                 |
> |                    | Peak Memory/MB | 14.58           | 7.06              | 3.39              | 2.85              | 7.42              | -                 |
>
> - Average per-epoch runtime and peak memory usage of IPAD:
>
> | Benchmark Datasets | Metrics     | $\mathcal{T}_0$ | $\mathcal{T}_{1}$ | $\mathcal{T}_{2}$ | $\mathcal{T}_{3}$ | $\mathcal{T}_{4}$ | $\mathcal{T}_{5}$ |
> |--------------------|----------------|-----------------|-------------------|-------------------|-------------------|-------------------|-------------------|
> | CS-CL              | Runtime/ms     | -               | 2.39              | 2.87              | 3.61              | 4.09              | 4.61              |
> |                    | Peak Memory/MB | -               | 1.72              | 2.40              | 3.09              | 4.19              | 4.62              |
> | Reddit-CL          | Runtime/s      | -               | 0.07              | 0.07              | 0.07              | 0.11              | -                 |
> |                    | Peak Memory/MB | -               | 56.45             | 62.72             | 62.77             | 111.22            | -                 |
>
> - Average per-epoch runtime and peak memory usage of DBP:
>
> | Benchmark Datasets | Metrics     | $\mathcal{T}_0$ | $\mathcal{T}_{1}$ | $\mathcal{T}_{2}$ | $\mathcal{T}_{3}$ | $\mathcal{T}_{4}$ | $\mathcal{T}_{5}$ |
> |--------------------|----------------|-----------------|-------------------|-------------------|-------------------|-------------------|-------------------|
> | CS-CL              | Runtime/ms     | 3.68            | 2.50              | 2.31              | 2.29              | 2.54              | 1.86              |
> |                    | Peak Memory/MB | 9.13            | 6.95              | 5.41              | 5.63              | 10.51             | 25.25             |
> | Reddit-CL          | Runtime/s      | 0.50             | 0.04              | 0.03              | 0.03              | 0.04              | -                 |
> |                    | Peak Memory/MB | 745.54          | 180.61            | 175.58            | 162.29            | 304.97            | -                 |

---

> ### Author Response · Authors · 2025-11-20
> **Responses to W1 (Part 3)**
>
> Furthermore, we investigated the scalability of the IPAD and DBP components by systematically varying the hyperparameters $S _ t$ and $K$, respectively, and measuring their average per-epoch runtime and peak memory footprint on the incremental tasks $\mathcal{T} _ {t>0}$. As shown in the table below, the findings reveal that increasing $\|S _ t\|$ or $K$ incurs only a marginal effect on runtime. Although peak memory utilization exhibits a moderate rise with larger hyperparameter values, the overhead remains entirely inconsequential relative to the 24GB capacity of an NVIDIA GeForce RTX 3090 GPU. Notably, FDC is a widely employed post-processing strategy in the literature rather than a central contribution of our framework; thus, a detailed scalability evaluation was not undertaken.
>
> | Benchmark Datasets | Metrics     | IPAD-$\|S _ t\|$ |        |        |  DBP-$K$ |        |        |
> |--------------------|----------------|:-----------:|:------:|:------:|:------:|:------:|:------:|
> |                    |                |     100     |   200  |   300  |   10   |   20   |   30   |
> | CS-CL              | Runtime/ms     |     3.52    |  3.79  |  4.19  |  2.30  |  2.37  |  1.93  |
> |                    | Peak Memory/MB |     3.20    |  6.52  |  9.83  |  10.75 |  18.19 |  26.27 |
> | Reddit-CL          | Runtime/s      |     0.08    |  0.09  |  0.09  |  0.04  |  0.04  |  0.05  |
> |                    | Peak Memory/MB |    73.29    | 143.78 | 214.69 | 205.86 | 387.09 | 561.10 |

---

> ### Author Response · Authors · 2025-11-20
> **Responses to W2**
>
> We thank the reviewer for the insightful comments.
>
> In fact, the derivation of Theorem 1 and the empirical study in Figure 1 do not involve IPAD or feature drift compensation; hence, the entire process is independent of the hyperparameters $\beta$ and $\gamma$. Furthermore, as you suggested, Theorem 1 could be refined to yield a tighter drift bound. However, even a relatively loose KL-based analysis suffices to effectively justify the rationale behind our motivation. While a stricter bound could be obtained under additional assumptions, it would not alter the qualitative trend of our theoretical conclusions.

---

> ### Author Response · Authors · 2025-11-20
> **Responses to W3**
>
> We thank the reviewer for the insightful comments.
>
> Following your suggestion, we assessed the effect of margin-, energy-, and entropy-based hard example selection strategies on the performance of our proposed IPAL across both the small-scale CS-CL and large-scale Reddit-CL benchmark datasets. Specifically, given a node in task $\mathcal{T} _ t$ with label $c$, its embedding is denoted as $f _ {\theta _ t}^c$.
>
> **Margin-based method [1].** We employ EL2N as a margin-oriented metric, which furnishes a reliable indicator of example hardness by estimating the gradient norm. Formally, EL2N is defined as:
>
> $$\mathrm{EL2N}(x _ t^c)=\|\|softmax({f _ {\theta _ t}^c}^{\mathrm{T}}\cdot\left[\mu _ k\right]_{k\in\mathcal{Y} _ {0:t}})-\mathrm{onehot}(c)\|\| _ 2,$$
>
> where $\left[\mu _ k\right]_{k\in\mathcal{Y} _ {0:t}}$ denotes the concatenation of offline and online prototypes, and $\mathrm{onehot}(c)$ represents the one-hot encoding of label $c$.
>
> **Energy-based method [2].** The free energy quantifies the likelihood of an input sample, with higher energy values indicating less favorable states for the model and, consequently, greater learning difficulty. The free energy is computed as:
>
> $$F(x_t^c)=-\mathrm{log}\sum _ {k\in\mathcal{Y} _ {0:t}}e^{-E(x _ t^c,k)},$$
>
> where $E(x_t^c,k)=-{f _ {\theta _ t}^c}^{\mathrm{T}}\cdot\left[\mu _ k\right] _ {k\in\mathcal{Y} _ {0:t}}$.
>
> **Entropy-based method.** Refer to Equation (9) in the paper.
>
> The table shows that hard examples selected by different metrics vary, influencing PCL’s learning dynamics. The margin-based strategy yields the most pronounced gains by incorporating explicit label information, while free-energy- and entropy-based strategies also improve performance. Removing the DBP mechanism substantially degrades performance, highlighting that hard examples, as surrogates for the decision boundary, enhance inter-class separability and bolster continual graph learning. Notably, the DBP selection procedure is not limited to these strategies and can be replaced with other state-of-the-art alternatives.
>
> | Methods   | CS-CL          |                | Reddit-CL      |                |
> |-----------|----------------|----------------|----------------|----------------|
> |           | AP/%$\uparrow$ | AF/%$\uparrow$ | AP/%$\uparrow$ | AF/%$\uparrow$ |
> | w/o DBP   | 75.94$\pm$3.05     | -21.05$\pm$4.00    | 88.08$\pm$1.01     | -4.45$\pm$1.65     |
> | w/Margin  | 86.20$\pm$2.34     | -8.60$\pm$2.25     | 93.38$\pm$0.28     | -0.40$\pm$0.14     |
> | w/Energy  | 79.81$\pm$2.14     | -16.54$\pm$1.69    | 90.92$\pm$1.03     | -0.80$\pm$0.32     |
> | w/Entropy | 83.07$\pm$2.16     | -12.89$\pm$2.50    | 92.15$\pm$0.13     | -0.27$\pm$0.19     |
>
> [1] Sabbineni A, Anand N, Minakova M. Comprehensive Benchmarking of Entropy and Margin Based Scoring Metrics for Data Selection[J]. arXiv preprint arXiv:2311.16302, 2023.
>
> [2] Zhang X, Chuah J H, Loo C K, et al. An Energy Sampling Replay-Based Continual Learning Framework[C]//International Conference on Artificial Neural Networks. Cham: Springer Nature Switzerland, 2024: 17-30.

---

### Official Review · Reviewer_C2Qe · 2025-10-30

**Soundness:** 3
**Presentation:** 3
**Contribution:** 2
**Rating:** 6
**Confidence:** 3

**Summary:**

The paper presents Instance-Prototype Affinity Learning (IPAL), a framework for Non-Exemplar Continual Graph Learning (NECGL) that mitigates catastrophic forgetting without storing prior samples. Built upon Prototype Contrastive Learning (PCL), IPAL introduces three key components: Topology-Integrated Gaussian Prototypes (TIGP) to incorporate graph topology via PageRank-based node importance, Instance-Prototype Affinity Distillation (IPAD) for flexible relation-based knowledge retention, and Decision Boundary Perception (DBP) to enhance class separability using boundary-aware samples. Experiments on four benchmark datasets demonstrate that IPAL consistently improves performance and achieves a better trade-off between stability and plasticity compared with recent state-of-the-art methods.

**Strengths:**

1. The paper addresses a meaningful challenge in non-exemplar continual graph learning where privacy and memory constraints prevent rehearsal.
2. The three modules (TIGP, IPAD, DBP) are well-motivated, mutually complementary, and grounded in clear intuition.
3. Extensive experiments across four benchmarks with ablation and sensitivity analyses support the effectiveness of each component.

**Weaknesses:**

1. The framework is an evolutionary extension of PCL and prototype replay ideas rather than a fundamentally new paradigm.
2. The paper lacks concrete analysis of computational cost, parameter overhead, or runtime efficiency.

**Questions:**

What is the relationship with the related work [1]?
[1] Chaoxi Niu, Guansong Pang, Ling Chen, and Bing Liu. Replay-and-forget-free graph classincremental learning: A task profiling and prompting approach. Advances in Neural Information
Processing Systems, 37:87978–88002, 2024b.

---

> ### Author Response · Authors · 2025-11-19
> **Responses to W1**
>
> We thank the reviewer for the insightful comments.
>
> Our work builds upon the existing PCL framework to propose a novel Non-Exemplar Continual Graph Learning approach. Empirically, we observed an interesting phenomenon: PCL exhibits less pronounced drift than PR, and we provide corresponding theoretical analysis to support this observation—a facet that remains unexplored in existing research. Motivated by this finding, we introduce three new components—TIGP, IPAD, and DBP—integrated into PCL to further enhance model plasticity and stability. These components seamlessly complement the PCL framework, effectively improving performance in continual graph learning. Our core contribution spans from empirical observation to theoretical justification and framework innovation, distinguishing our approach markedly from existing PCL methods.

---

> ### Author Response · Authors · 2025-11-19
> **Responses to W2 (Part 1)**
>
> We thank the reviewer for the insightful comments. Following your suggestion, we evaluated the efficiency of the framework and its constituent components, including PageRank, TIGP, IPAD, and DBP.
>
> **PageRank:** First, we investigated the time complexity of the PageRank algorithm. For task $\mathcal{T} _ t$, the computational cost is primarily dominated by the matrix multiplication between the transition matrix $\mathbf{P} _ t=\mathbf{A} _ t^{\mathrm{T}}\mathbf{D}^{-1}\in\mathbb{R}^{|\mathcal{V} _ t|\times|\mathcal{V} _ t|}$ and the PageRank vector $\mathbf{R} _ t\in\mathbb{R}^{|\mathcal{V} _ t|}$. When $\mathbf{P} _ t$ is treated as a dense matrix, $n$ iterations incur a time complexity of $\mathcal{O}(n|\mathcal{V} _ t|^2)$; leveraging the sparsity of $\mathbf{P} _ t$ reduces the time complexity to $\mathcal{O}(n|\mathcal{E} _ t|)$. Furthermore, we evaluated the computational efficiency of PageRank on both the small-scale CS-CL and the large-scale Reddit-CL benchmark datasets, considering dense matrix computations. PageRank on CS-CL was executed on a single NVIDIA GeForce RTX 3090 GPU, whereas due to out-of-memory (OOM) issues with Reddit-CL, computations were performed on a CPU (Intel Xeon Gold 6226R, 64 cores, 2.90 GHz) with 251 GiB of RAM. Experimental results are shown below. On the small-scale benchmark dataset CS-CL, the computational cost of PageRank is on the order of microseconds and is negligible during training. For the large-scale benchmark Reddit-CL, as the base task $\mathcal{T} _ 0$ involves nodes from 20 classes and their interconnecting edges, the PageRank computation incurs a slightly higher cost; yet for incremental tasks $\mathcal{T}_{t>0}$, it remains only a few seconds. Crucially, PageRank can be precomputed prior to each task, allowing the use of the resulting $\mathbf{R}_t$ during training without any additional overhead.
>
> | Benchmark Datasets | Info.      | $\mathcal{T}_0$ | $\mathcal{T}_1$ | $\mathcal{T}_2$ | $\mathcal{T}_3$ | $\mathcal{T}_4$ | $\mathcal{T}_5$ |
> |--------------------|------------|-----------------|-----------------|-----------------|-----------------|-----------------|-----------------|
> | CS-CL              | # nodes    | 5043            | 2564            | 1699            | 1562            | 2453            | 5012            |
> |                    | # edges    | 39884           | 16602           | 14000           | 13686           | 16576           | 39148           |
> |                    | Runtime/ms | 8.86            | 5.09            | 4.83            | 4.29            | 5.08            | 8.29            |
> | Reddit-CL          | # nodes    | 133970          | 26358           | 21634           | 17380           | 28511           | -               |
> |                    | # edges    | 56195430        | 5403916         | 4637618         | 10989194        | 15995550        | -               |
> |                    | Runtime/s  | 73.31           | 4.06            | 2.41            | 1.40            | 4.22            | -               |

---

> ### Author Response · Authors · 2025-11-19
> **Responses to W2 (Part 2)**
>
> We next report the average per-epoch runtime and peak memory usage of the proposed TIGP, IPAD, and DBP components on CS-CL and Reddit-CL. Notably, TIGP here involves only the computation of online prototypes, while DBP includes both hard example retrieval and PCL loss computation.
>
> All experiments were conducted on a single NVIDIA GeForce RTX 3090 GPU. The three tables below report the task-level average per-epoch runtime and peak memory usage. On the small-scale CS-CL benchmark, each component exhibits microsecond-level runtimes and low memory consumption. On the large-scale Reddit-CL benchmark, the base task $\mathcal{T} _ 0$, which contains an order of magnitude more nodes and edges than incremental tasks $\mathcal{T} _ {t>0}$, incurs higher computational and memory costs, while the overhead for incremental tasks remains substantially lower. Notably, all reported runtimes and memory usage are fully manageable on a single 24GB RTX 3090 GPU.
>
> - Average per-epoch runtime and peak memory usage of TIGP:
>
> | Benchmark Datasets | Metrics     | $\mathcal{T}_0$ | $\mathcal{T}_{1}$ | $\mathcal{T}_{2}$ | $\mathcal{T}_{3}$ | $\mathcal{T}_{4}$ | $\mathcal{T}_{5}$ |
> |--------------------|----------------|-----------------|-------------------|-------------------|-------------------|-------------------|-------------------|
> | CS-CL              | Runtime/ms     | 1.10             | 0.73              | 0.53              | 0.55              | 0.55              | 0.53              |
> |                    | Peak Memory/MB | 2.47            | 2.99              | 0.54              | 0.85              | 3.10              | 5.57              |
> | Reddit-CL          | Runtime/s      | 0.24            | 0.01              | 0.01              | 0.01              | 0.01              | -                 |
> |                    | Peak Memory/MB | 14.58           | 7.06              | 3.39              | 2.85              | 7.42              | -                 |
>
> - Average per-epoch runtime and peak memory usage of IPAD:
>
> | Benchmark Datasets | Metrics     | $\mathcal{T}_0$ | $\mathcal{T}_{1}$ | $\mathcal{T}_{2}$ | $\mathcal{T}_{3}$ | $\mathcal{T}_{4}$ | $\mathcal{T}_{5}$ |
> |--------------------|----------------|-----------------|-------------------|-------------------|-------------------|-------------------|-------------------|
> | CS-CL              | Runtime/ms     | -               | 2.39              | 2.87              | 3.61              | 4.09              | 4.61              |
> |                    | Peak Memory/MB | -               | 1.72              | 2.40              | 3.09              | 4.19              | 4.62              |
> | Reddit-CL          | Runtime/s      | -               | 0.07              | 0.07              | 0.07              | 0.11              | -                 |
> |                    | Peak Memory/MB | -               | 56.45             | 62.72             | 62.77             | 111.22            | -                 |
>
> - Average per-epoch runtime and peak memory usage of DBP:
>
> | Benchmark Datasets | Metrics     | $\mathcal{T}_0$ | $\mathcal{T}_{1}$ | $\mathcal{T}_{2}$ | $\mathcal{T}_{3}$ | $\mathcal{T}_{4}$ | $\mathcal{T}_{5}$ |
> |--------------------|----------------|-----------------|-------------------|-------------------|-------------------|-------------------|-------------------|
> | CS-CL              | Runtime/ms     | 3.68            | 2.50              | 2.31              | 2.29              | 2.54              | 1.86              |
> |                    | Peak Memory/MB | 9.13            | 6.95              | 5.41              | 5.63              | 10.51             | 25.25             |
> | Reddit-CL          | Runtime/s      | 0.50             | 0.04              | 0.03              | 0.03              | 0.04              | -                 |
> |                    | Peak Memory/MB | 745.54          | 180.61            | 175.58            | 162.29            | 304.97            | -                 |

---

> ### Author Response · Authors · 2025-11-19
> **Responses to W2 (Part 3)**
>
> Furthermore, we compared the average per-epoch training time and peak memory usage of our proposed IPAL with the state-of-the-art PR methods POLO [1] and EFC [2] across both benchmark datasets. The results in the table below show that IPAL attains comparable or superior efficiency to EFC in both training time and memory consumption, while substantially outperforming POLO.
>
> | Benchmark Datasets | Methods | $\mathcal{T}_0$ |                | $\mathcal{T}_{1}$ |                | $\mathcal{T}_{2}$ |                | $\mathcal{T}_{3}$ |                | $\mathcal{T}_{4}$ |                | $\mathcal{T}_{5}$ |                |
> |--------------------|---------|-----------------|----------------|-------------------|----------------|-------------------|----------------|-------------------|----------------|-------------------|----------------|-------------------|----------------|
> |                    |         | Training Time/s | Peak Memory/MB | Training Time/s         | Peak Memory/MB | Training Time/s         | Peak Memory/MB | Training Time/s         | Peak Memory/MB | Training Time/s         | Peak Memory/MB | Training Time/s         | Peak Memory/MB |
> | CS-CL              | POLO    | 0.01            | 300.10         | 0.07              | 158.99         | 0.10              | 124.31         | 0.11              | 119.13         | 0.12              | 155.69         | 0.14              | 305.49         |
> |                    | EFC     | 0.01            | 301.88         | 0.02              | 160.56         | 0.02              | 125.56         | 0.02              | 120.44         | 0.02              | 158.90         | 0.03              | 307.89         |
> |                    | IPAL    | 0.02            | 300.81         | 0.02              | 160.21         | 0.02              | 125.23         | 0.02              | 119.74         | 0.03              | 156.61         | 0.03              | 306.62         |
> | Reddit-CL          | POLO    | 5.31            | 1483.81        | 1.86              | 616.96         | 1.85              | 588.23         | 1.74              | 499.49         | 3.01              | 698.72         | -                 | -              |
> |                    | EFC     | 5.08            | 1463.27        | 0.62              | 721.59         | 0.53              | 686.50         | 0.45              | 577.37         | 0.79              | 820.20         | -                 | -              |
> |                    | IPAL    | 5.86            | 1059.96        | 0.57              | 425.54         | 0.48              | 403.87         | 0.39              | 340.07         | 0.72              | 482.59         | -                 | -              |
>
> [1] Wang S, Shi W, He Y, et al. Non-exemplar class-incremental learning via adaptive old class reconstruction[C]//Proceedings of the 31st ACM International Conference on Multimedia. 2023: 4524-4534.
>
> [2] Magistri S, Trinci T, Soutif-Cormerais A, et al. Elastic feature consolidation for cold start exemplar-free incremental learning[J]. arXiv preprint arXiv:2402.03917, 2024.

---

> ### Author Response · Authors · 2025-11-19
> **Responses to Q1**
>
> We thank the reviewer for the insightful comments.
>
> TPP belongs to the family of parameter-isolation approaches, which alleviate inter-task interference by assigning task-specific graph prompts and classification heads. Its core mechanism hinges on identifying the task ID through Laplacian-smoothed task-prototype matching, after which the corresponding prompts and heads are activated to enable class-incremental prediction. In essence, this paradigm instantiates an independent, non-shared parameter space for each task. In contrast, the proposed IPAL framework operates under a unified, parameter-sharing architecture across the entire task stream, explicitly addressing the pervasive feature drift problem characteristic of the exemplar-free setting. Within this paradigm, class prototypes serve as surrogates for historical class distributions and are explicitly leveraged through replay-based mechanisms to counteract catastrophic forgetting. TPP, however, utilizes task prototypes exclusively for task-ID inference. Moreover, existing research [1] has shown that TPP may suffer from task-ID leakage, and for fairness, we therefore do not include it in our comparisons. We have also appropriately cited this work in the Related Work section.
>
> [1] Cheng Z, Li Z, Li Y, et al. Can LLMs Alleviate Catastrophic Forgetting in Graph Continual Learning? A Systematic Study[J]. arXiv preprint arXiv:2505.18697, 2025.

---

### Official Review · Reviewer_Zp24 · 2025-10-31

**Soundness:** 3
**Presentation:** 3
**Contribution:** 3
**Rating:** 6
**Confidence:** 3

**Summary:**

The paper studies the problem of Non-Exemplar Continual Graph Learning (NECGL), where models must learn new graph tasks without storing past data. The authors build upon Prototype Contrastive Learning (PCL) and propose a new framework, Instance-Prototype Affinity Learning (IPAL), to alleviate feature drift and catastrophic forgetting. IPAL introduces three modules: Topology-Integrated Gaussian Prototypes (TIGP) for topology-aware prototypes, Instance-Prototype Affinity Distillation (IPAD) for flexible knowledge retention, and Decision Boundary Perception (DBP) for better class separation. Experiments on four benchmark datasets demonstrate consistent improvements over existing Non-Exemplar methods, suggesting that IPAL achieves a better balance between stability and plasticity.

**Strengths:**

1. The use of PageRank-based topology integration (TIGP) is a meaningful attempt to incorporate graph structural importance into prototype construction, improving representation of high-impact nodes.

2. The proposed Instance-Prototype Affinity Distillation (IPAD) provides a more flexible alternative to traditional feature distillation, maintaining inter- and intra-class relations without over-constraining the feature space.

3.The Decision Boundary Perception (DBP) mechanism is a thoughtful addition that leverages high-entropy (hard) samples near class boundaries to enhance inter-class separation.

**Weaknesses:**

1. It is not entirely clear why PageRank-weighted nodes would yield better class prototypes, as the paper provides limited explanation or analysis for this design choice. High-centrality nodes may not adequately represent peripheral or low-degree nodes, potentially biasing the prototypes toward graph centers.

2. The framework introduces several additional components but does not report training time or memory usage. It remains unclear whether these modules introduce notable computational or memory overhead.

3. While the paper includes some parameter analysis, it remains unclear how much re-tuning would be required when applying IPAL to new datasets. The current experiments focus on a fixed set of benchmarks, and it would be helpful to clarify whether the same hyperparameter settings generalize well or if dataset-specific tuning is necessary.

**Questions:**

1. Could the authors provide more explanation or empirical evidence on why PageRank weighting leads to better prototype representations? Have they compared it with other node-importance measures (e.g., degree centrality or attention-based scores)?

2. How significant is the computational overhead introduced by PageRank computation, Mixup synthesis, and drift compensation? Any quantitative report on training time or GPU memory usage would be helpful.

3. Regarding hyperparameters (γ, β, K, |Sₜ|), could the authors clarify whether the same configuration works well across all datasets, or if substantial re-tuning is needed for different graph domains?

---

> ### Author Response · Authors · 2025-11-19
> **Responses to W1 and Q1**
>
> We thank the reviewer for the insightful comments.
>
> Given that nodes reside in markedly different local topological configurations, their contributions to the global data distribution should not be treated uniformly. Class prototypes, which function as surrogates for the underlying class distributions, benefit from the incorporation of PageRank-based topological weighting, enabling more faithful estimation of node influence and, consequently, more representative prototype construction. Within the PCL framework, the fidelity of these prototypes is critical to overall performance. Although peripheral nodes or low-degree nodes theoretically exert limited influence on their class distributions, DBP deliberately identifies such boundary-prone instances to enhance inter-class separability and prevent ambiguity during incremental learning. Following your suggestion, we compared the PageRank-based weighting with degree centrality weighting on the small-scale CS-CL and large-scale Reddit-CL benchmark datasets. It can be observed that the PageRank-based approach adopted in this work significantly outperforms the degree centrality method. On one hand, degree centrality considers only local neighborhood structures, whereas PageRank, through random walks, more effectively captures global topological information. On the other hand, in the presence of isolated nodes within the task graph, degree centrality fails to assign meaningful influence (always yielding zero), while PageRank leverages the damping factor $\alpha$ to distribute more reasonable influence weights.
>
> | Methods           | CS-CL          |                | Reddit-CL      |                |
> |-------------------|----------------|----------------|----------------|----------------|
> |                   | AP/%$\uparrow$ | AF/%$\uparrow$ | AP/%$\uparrow$ | AF/%$\uparrow$ |
> | Degree Centrality | 80.16$\pm$2.00 | -15.95$\pm$2.10    | 91.60$\pm$0.35     | -0.77$\pm$0.38     |
> | PageRank          | 83.07$\pm$2.16 | -12.89$\pm$2.50    | 92.15$\pm$0.13     | -0.27$\pm$0.19     |

---

> ### Author Response · Authors · 2025-11-19
> **Responses to W2 and Q2 (Part 1)**
>
> We thank the reviewer for the insightful comments. Following your suggestion, we assessed the efficiency of the proposed components.
>
> **PageRank:** First, we investigated the time complexity of the PageRank algorithm. For task $\mathcal{T} _ t$, the computational cost is primarily dominated by the matrix multiplication between the transition matrix $\mathbf{P} _ t=\mathbf{A} _ t^{\mathrm{T}}\mathbf{D}^{-1}\in\mathbb{R}^{|\mathcal{V} _ t|\times|\mathcal{V} _ t|}$ and the PageRank vector $\mathbf{R} _ t\in\mathbb{R}^{|\mathcal{V} _ t|}$. When $\mathbf{P} _ t$ is treated as a dense matrix, $n$ iterations incur a time complexity of $\mathcal{O}(n|\mathcal{V} _ t|^2)$; leveraging the sparsity of $\mathbf{P} _ t$ reduces the time complexity to $\mathcal{O}(n|\mathcal{E} _ t|)$. Furthermore, we evaluated the computational efficiency of PageRank on both the small-scale CS-CL and the large-scale Reddit-CL benchmark datasets, considering dense matrix computations. PageRank on CS-CL was executed on a single NVIDIA GeForce RTX 3090 GPU, whereas due to out-of-memory (OOM) issues with Reddit-CL, computations were performed on a CPU (Intel Xeon Gold 6226R, 64 cores, 2.90 GHz) with 251 GiB of RAM. Experimental results are shown below. On the small-scale benchmark dataset CS-CL, the computational cost of PageRank is on the order of microseconds and is negligible during training. For the large-scale benchmark Reddit-CL, as the base task $\mathcal{T} _ 0$ involves nodes from 20 classes and their interconnecting edges, the PageRank computation incurs a slightly higher cost; yet for incremental tasks $\mathcal{T}_{t>0}$, it remains only a few seconds. Crucially, PageRank can be precomputed prior to each task, allowing the use of the resulting $\mathbf{R}_t$ during training without any additional overhead.
>
> | Benchmark Datasets | Info.      | $\mathcal{T}_0$ | $\mathcal{T}_1$ | $\mathcal{T}_2$ | $\mathcal{T}_3$ | $\mathcal{T}_4$ | $\mathcal{T}_5$ |
> |--------------------|------------|-----------------|-----------------|-----------------|-----------------|-----------------|-----------------|
> | CS-CL              | # nodes    | 5043            | 2564            | 1699            | 1562            | 2453            | 5012            |
> |                    | # edges    | 39884           | 16602           | 14000           | 13686           | 16576           | 39148           |
> |                    | Runtime/ms | 8.86            | 5.09            | 4.83            | 4.29            | 5.08            | 8.29            |
> | Reddit-CL          | # nodes    | 133970          | 26358           | 21634           | 17380           | 28511           | -               |
> |                    | # edges    | 56195430        | 5403916         | 4637618         | 10989194        | 15995550        | -               |
> |                    | Runtime/s  | 73.31           | 4.06            | 2.41            | 1.40            | 4.22            | -               |

---

> ### Author Response · Authors · 2025-11-19
> **Responses to W2 and Q2 (Part 2)**
>
> We next report the average per-epoch runtime and peak memory usage of the proposed TIGP, IPAD, and DBP components on CS-CL and Reddit-CL. Notably, TIGP here involves only the computation of online prototypes, while DBP includes both hard example retrieval and PCL loss computation. Moreover, FDC is a commonly used post-processing strategy in existing studies and does not constitute a core contribution of our work; therefore, evaluating its efficiency is not essential.
>
> All experiments were conducted on a single NVIDIA GeForce RTX 3090 GPU. The three tables below report the task-level average per-epoch runtime and peak memory usage. On the small-scale CS-CL benchmark, each component exhibits microsecond-level runtimes and low memory consumption. On the large-scale Reddit-CL benchmark, the base task $\mathcal{T} _ 0$, which contains an order of magnitude more nodes and edges than incremental tasks $\mathcal{T} _ {t>0}$, incurs higher computational and memory costs, while the overhead for incremental tasks remains substantially lower. Notably, all reported runtimes and memory usage are fully manageable on a single 24GB RTX 3090 GPU.
>
> - Average per-epoch runtime and peak memory usage of TIGP:
>
> | Benchmark Datasets | Metrics     | $\mathcal{T}_0$ | $\mathcal{T}_{1}$ | $\mathcal{T}_{2}$ | $\mathcal{T}_{3}$ | $\mathcal{T}_{4}$ | $\mathcal{T}_{5}$ |
> |--------------------|----------------|-----------------|-------------------|-------------------|-------------------|-------------------|-------------------|
> | CS-CL              | Runtime/ms     | 1.10             | 0.73              | 0.53              | 0.55              | 0.55              | 0.53              |
> |                    | Peak Memory/MB | 2.47            | 2.99              | 0.54              | 0.85              | 3.10              | 5.57              |
> | Reddit-CL          | Runtime/s      | 0.24            | 0.01              | 0.01              | 0.01              | 0.01              | -                 |
> |                    | Peak Memory/MB | 14.58           | 7.06              | 3.39              | 2.85              | 7.42              | -                 |
>
> - Average per-epoch runtime and peak memory usage of IPAD:
>
> | Benchmark Datasets | Metrics     | $\mathcal{T}_0$ | $\mathcal{T}_{1}$ | $\mathcal{T}_{2}$ | $\mathcal{T}_{3}$ | $\mathcal{T}_{4}$ | $\mathcal{T}_{5}$ |
> |--------------------|----------------|-----------------|-------------------|-------------------|-------------------|-------------------|-------------------|
> | CS-CL              | Runtime/ms     | -               | 2.39              | 2.87              | 3.61              | 4.09              | 4.61              |
> |                    | Peak Memory/MB | -               | 1.72              | 2.40              | 3.09              | 4.19              | 4.62              |
> | Reddit-CL          | Runtime/s      | -               | 0.07              | 0.07              | 0.07              | 0.11              | -                 |
> |                    | Peak Memory/MB | -               | 56.45             | 62.72             | 62.77             | 111.22            | -                 |
>
> - Average per-epoch runtime and peak memory usage of DBP:
>
> | Benchmark Datasets | Metrics     | $\mathcal{T}_0$ | $\mathcal{T}_{1}$ | $\mathcal{T}_{2}$ | $\mathcal{T}_{3}$ | $\mathcal{T}_{4}$ | $\mathcal{T}_{5}$ |
> |--------------------|----------------|-----------------|-------------------|-------------------|-------------------|-------------------|-------------------|
> | CS-CL              | Runtime/ms     | 3.68            | 2.50              | 2.31              | 2.29              | 2.54              | 1.86              |
> |                    | Peak Memory/MB | 9.13            | 6.95              | 5.41              | 5.63              | 10.51             | 25.25             |
> | Reddit-CL          | Runtime/s      | 0.50             | 0.04              | 0.03              | 0.03              | 0.04              | -                 |
> |                    | Peak Memory/MB | 745.54          | 180.61            | 175.58            | 162.29            | 304.97            | -                 |

---

> ### Author Response · Authors · 2025-11-19
> **Responses to W3 and Q3**
>
> We thank the reviewer for the insightful comments.
>
> In fact, the issues you mentioned have already been analyzed in Section 5.4 and Appendix C. In this work, we adopt a unified setting of $|S _ t|=100$, $K=10$, and $\beta=0.1$ across all four datasets, which consistently yields strong performance. In continual learning, balancing model plasticity and stability remains a fundamental challenge. The hyperparameter $\gamma$ regulates this trade-off, necessitating dataset-specific tuning. Nevertheless, as shown in Figure 5, a value around 0.7 achieves a favorable balance between plasticity and stability, consistently delivering superior performance on all datasets while streamlining the hyperparameter tuning process.

---

### Official Review · Reviewer_Rbf5 · 2025-11-01

**Soundness:** 3
**Presentation:** 3
**Contribution:** 2
**Rating:** 4
**Confidence:** 3

**Summary:**

The paper tackles non-exemplar continual graph learning (NECGL)—updating a GNN over a sequence of class-incremental tasks when no past examples may be stored. It observes that the usual prototype replay (PR) with cross-entropy suffers from feature drift as the encoder changes over tasks, and shows that prototype contrastive learning (PCL) drifts less, backed by a KL-divergence analysis and visual evidence. Building on that, the authors propose IPAL (Instance-Prototype Affinity Learning): a PCL-based framework that injects graph topology into prototypes, distils relations between instances and prototypes, and sharpens decision boundaries using hard examples. Across four node-classification benchmarks, IPAL improves average performance while maintaining a good stability–plasticity trade-off.

**Strengths:**

1. Sound theoretical footing for using PCL in NECGL:
The authors formalize “feature drift” and prove (via a KL-divergence analysis under Gaussian assumptions) that Prototype Contrastive Learning (PCL) incurs strictly less drift than conventional Prototype Replay (PR). This gives a clear, principled reason to prefer PCL in non-exemplar continual graph learning, not just an empirical hunch.

2. Topology-aware prototypes that actually use the graph:
The Topology-Integrated Gaussian Prototypes (TIGP) weight class statistics by PageRank, so influential nodes shape the prototype more than peripheral ones, an intuitively correct use of graph structure that plain mean prototypes ignore. The paper also notes PageRank is computed once per task, keeping overhead low.

3. Relation-based distillation that preserves plasticity:
The Instance–Prototype Affinity Distillation (IPAD) regularizes instance and prototype relations instead of full feature vectors. The paper argues and illustrates that classic feature distillation can over-constrain the encoder and impede learning new classes, whereas IPAD aligns naturally with PCL and avoids that rigidity. This is a concrete, easy-to-adopt design change with clear motivation.

4. Explicit boundary sharpening to tackle single-prototype ambiguity:
The Decision Boundary Perception (DBP) module identifies high-entropy  nodes near boundaries and folds them into the PCL objective, directly encouraging inter-class separation, a targeted fix for ambiguity when one prototype can’t capture class diversity.

**Weaknesses:**

1. Theory relies on restrictive assumptions and local approximations:
The “PCL drifts less than PR” claim is proved under multivariate Gaussian feature distributions with positive-definite covariances, and the derivation uses a first-order Taylor approximation of the encoder’s update (i.e., infinitesimal step). That makes the result sensitive to non-Gaussian embeddings and larger optimization steps typical in practice. Clarifying the conditions (e.g., step size, τ, optimizer) under which the strict inequality provably holds—or adding stress tests for non-Gaussian / heavy-tailed features—would strengthen the claim.
2. Computational overhead is asserted, not quantified:
The method claims “PageRank is computed once per task … imposing no extra burden”, but there’s no runtime or asymptotic analysis versus baselines, it’s important for large graphs where PageRank and per-class covariance can still be costly. Please report training time, memory, and scaling curves.
3. Single-prototype limitation is acknowledged but not fully addressed:
The paper itself notes that a single class prototype can be inadequate and causes inter-class ambiguity; DBP then adds hard instance mining yet still keeps one prototype per class. A more direct fix (e.g., multi-prototype / mixture per class or topology-aware subclusters) is not explored. Including a comparison to a multi-prototype variant would test whether DBP fully compensates for intra-class diversity.

**Questions:**

Q1. Theory scope & assumptions:
What concrete conditions (on feature distributions, optimizer/step size, temperature τ) are required for Theorem 1 to hold in practice, and how sensitive is the result beyond Gaussian features and first-order (infinitesimal-step) approximations?

Q2. Runtime & scalability claims:
You state PageRank is computed once per task with “no extra burden on training.” Could you provide wall-clock time, memory, and asymptotic complexity on the largest graph (Reddit-CL), and compare to PR baselines?

---

> ### Author Response · Authors · 2025-11-19
> **Responses to W1 and Q1**
>
> We thank the reviewer for the insightful comments.
>
> The proof of Theorem 1 is derived under the assumptions of a strict multivariate Gaussian distribution and a first-order Taylor expansion. Although these assumptions deviate from practical conditions, such idealized theoretical settings help clarify our motivation—namely, the distinct behaviors of PCL and PR with respect to feature drift. More importantly, the multivariate Gaussian assumption and the infinitesimal update step are standard premises widely adopted in existing continual learning theory [1–7]. Therefore, the assumptions in Theorem 1 are not exceptions but rather consensus foundations in this line of theoretical analysis. If these assumptions were entirely removed, the theoretical results of these studies would no longer hold.
>
> [1] Zenke F, Poole B, Ganguli S. Continual learning through synaptic intelligence[C]//International conference on machine learning. PMLR, 2017: 3987-3995.
>
> [2] Aljundi R, Babiloni F, Elhoseiny M, et al. Memory aware synapses: Learning what (not) to forget[C]//Proceedings of the European conference on computer vision (ECCV). 2018: 139-154.
>
> [3] Zhu F, Cheng Z, Zhang X Y, et al. Class-incremental learning via dual augmentation[J]. Advances in neural information processing systems, 2021, 34: 14306-14318.
>
> [4] Wang Z, Li Y, Shen L, et al. A unified and general framework for continual learning[J]. arXiv preprint arXiv:2403.13249, 2024.
>
> [5] Ren Y, Ke L, Li D, et al. Incremental graph classification by class prototype construction and augmentation[C]//Proceedings of the 32nd ACM International Conference on Information and Knowledge Management. 2023: 2136-2145.
>
> [6] Wang S, Shi W, He Y, et al. Non-exemplar class-incremental learning via adaptive old class reconstruction[C]//Proceedings of the 31st ACM International Conference on Multimedia. 2023: 4524-4534.
>
> [7] Magistri S, Trinci T, Soutif-Cormerais A, et al. Elastic feature consolidation for cold start exemplar-free incremental learning[J]. arXiv preprint arXiv:2402.03917, 2024.

---

> ### Author Response · Authors · 2025-11-19
> **Responses to W2 and Q2**
>
> We thank the reviewer for the insightful comments.
>
> Following your suggestion, we investigated the time complexity of the PageRank algorithm. For task $\mathcal{T}_t$, the computational cost is primarily dominated by the matrix multiplication between the transition matrix $\mathbf{P}_t=\mathbf{A}_t^{\mathrm{T}}\mathbf{D}^{-1}\in\mathbb{R}^{|\mathcal{V}_t|\times|\mathcal{V}_t|}$ and the PageRank vector $\mathbf{R}_t\in\mathbb{R}^{|\mathcal{V}_t|}$. When $\mathbf{P}_t$ is treated as a dense matrix, $n$ iterations incur a time complexity of $\mathcal{O}(n|\mathcal{V}_t|^2)$; leveraging the sparsity of $\mathbf{P}_t$ reduces the time complexity to $\mathcal{O}(n|\mathcal{E}_t|)$.
>
> Furthermore, we evaluated the computational efficiency of PageRank on both the small-scale CS-CL and the large-scale Reddit-CL benchmark datasets, considering dense matrix computations. PageRank on CS-CL was executed on a single NVIDIA GeForce RTX 3090 GPU, whereas due to out-of-memory (OOM) issues with Reddit-CL, computations were performed on a CPU (Intel Xeon Gold 6226R, 64 cores, 2.90 GHz) with 251 GiB of RAM. Experimental results are shown below. On the small-scale benchmark dataset CS-CL, the computational cost of PageRank is on the order of microseconds and is negligible during training. For the large-scale benchmark Reddit-CL, as the base task $\mathcal{T} _ 0$ involves nodes from 20 classes and their interconnecting edges, the PageRank computation incurs a slightly higher cost; yet for incremental tasks $\mathcal{T}_{t>0}$, it remains only a few seconds. Crucially, PageRank can be precomputed prior to each task, allowing the use of the resulting $\mathbf{R}_t$ during training without any additional overhead.
>
> | Benchmark Datasets | Info.      | $\mathcal{T}_0$ | $\mathcal{T}_1$ | $\mathcal{T}_2$ | $\mathcal{T}_3$ | $\mathcal{T}_4$ | $\mathcal{T}_5$ |
> |--------------------|------------|-----------------|-----------------|-----------------|-----------------|-----------------|-----------------|
> | CS-CL              | # nodes    | 5043            | 2564            | 1699            | 1562            | 2453            | 5012            |
> |                    | # edges    | 39884           | 16602           | 14000           | 13686           | 16576           | 39148           |
> |                    | Runtime/ms | 8.86            | 5.09            | 4.83            | 4.29            | 5.08            | 8.29            |
> | Reddit-CL          | # nodes    | 133970          | 26358           | 21634           | 17380           | 28511           | -               |
> |                    | # edges    | 56195430        | 5403916         | 4637618         | 10989194        | 15995550        | -               |
> |                    | Runtime/s  | 73.31           | 4.06            | 2.41            | 1.40            | 4.22            | -               |

---

> ### Author Response · Authors · 2025-11-19
> **Responses to W3**
>
> We thank the reviewer for the insightful comments.
>
> Following your suggestion, we developed a multi-prototype variant $\mathrm{IPAL}^{\dagger}$, replacing the online prototypes and DBP-retrieved hard examples with the online KMeans cluster centers $\mathcal{C}$. To ensure a fair comparison, the number of cluster centers per class was set to $|\mathcal{C}|=11$, matching the total number of IPAL’s online prototypes and hard examples (1$+$10 per class). Experiments were conducted on both the small-scale CS-CL and large-scale Reddit-CL benchmark datasets, as shown in the table below. It can be observed that $\mathrm{IPAL}$ substantially outperforms its multi-prototype variant on both benchmark datasets. Although using multiple prototypes can better characterize intra-class diversity than a single prototype, it remains insufficient in mitigating inter-class ambiguity. By contrast, the proposed DBP explicitly retrieves hard examples near class boundaries, thereby enhancing the separability of class distributions throughout incremental learning and ultimately improving overall performance in continual graph learning.
>
> | Methods                                          | CS-CL            |                  | Reddit-CL        |                  |
> |--------------------------------------------------|------------------|------------------|------------------|------------------|
> |                                                  | AP/%$\uparrow$ | AF/%$\uparrow$ | AP/%$\uparrow$ | AF/%$\uparrow$ |
> | $\mathrm{IPAL}^{\dagger}_{\|\mathcal{C}\|=11}$ | 82.08$\pm$1.58       | -14.34$\pm$1.75      | 91.38$\pm$0.99       | -3.20$\pm$1.28       |
> | $\mathrm{IPAL}$                                | 83.07$\pm$2.16       | -12.89$\pm$2.50      | 92.15$\pm$0.13       | -0.27$\pm$0.19       |

---

### Author Response · Authors · 2025-11-27
**Inquiry About Review Progress**

Dear Area Chairs and Program Chairs,

Thank you for your insightful suggestions on our paper. Following your suggestions, we have responded to the relevant points and revised the PDF accordingly. With only one week remaining in the rebuttal phase, I would like to kindly check whether there are any updates from the reviewers or if any further information from our side would be helpful. Your feedback will be invaluable in refining our paper.

Best regards,

The Authors

---

### Author Response · Authors · 2025-12-01
**Summary of Contributions**

Dear Area Chairs,

To facilitate a clearer understanding of our work and rebuttal, we provide a concise summary of our contributions.

✨ **Main Contributions**

Current non-exemplar continual graph learning (NECGL) methods generally rely on prototype replay to avoid memory and privacy concerns, yet their performance is fundamentally limited by feature drift. **Through empirical and theoretical analysis, we show that prototype contrastive learning (PCL) induces substantially lighter feature drift than prototype replay.** Building on this insight, we address several key limitations in existing NECGL approaches.

Although prototype replay and feature distillation are widely used to maintain model stability, three issues persist:
- Averaging-based prototype construction is isotropic and overlooks node importance shaped by graph topology.
- Feature distillation over-constrains the feature space, impairing model plasticity.
- A single prototype fails to capture class distributions, leading to inter-class ambiguity under PCL.

To overcome these issues, we propose **Instance-Prototype Affinity Learning (IPAL)** for NECGL. We compute node impact via PageRank and construct **Topology-Integrated Gaussian Prototypes (TIGP)** to better guide class distributions toward high-impact nodes. To mitigate forgetting without imposing excessive regularization, we introduce **Instance-Prototype Affinity Distillation (IPAD)**, which flexibly constrains the feature space via instance–prototype affinity alignment and integrates seamlessly with PCL. Furthermore, we incorporate a **Decision Boundary Perception (DBP)** mechanism to sharpen inter-class separation by repelling boundary-adjacent instances. Evaluations on four node classification benchmark datasets demonstrate that our method outperforms existing state-of-the-art approaches, achieving a superior trade-off between plasticity and stability.

✨ **Summary of Rebuttal Responses**

- Since all four reviewers raised concerns regarding the computational efficiency of the proposed method, we conducted additional evaluations of each component, including runtime, peak memory usage, and scalability. We also compared the overall efficiency of IPAL against state-of-the-art prototype replay methods such as POLO and EFC.
- Reviewers Rbf5 and KRRQ questioned the strict validity conditions of Theorem 1. In response, we provided a detailed explanation of its underlying principles, along with the necessary supporting materials.
- Following Reviewer Rbf5’s suggestion to compare DBP with multi-prototype variants, we implemented an online KMeans–based multi-prototype version and conducted a thorough comparison with DBP.
- Reviewer Zp24 recommended further explanation of the PageRank-weighted class prototypes and comparison with degree centrality. We included the corresponding experiments and analyses in the response.
- Reviewer Zp24 also expressed concerns regarding whether a uniform set of hyperparameters can generalize across datasets. We clarified this point by referring to the parameter studies provided in the paper and elaborating on the observed robustness.
- Reviewer C2Qe questioned the novelty of our method. We clarified our contributions and highlighted their distinctions from prior work.
- Reviewer KRRQ suggested evaluating different hard-example selection strategies for DBP. Accordingly, we incorporated margin-based and energy-based selection methods and demonstrated that DBP remains compatible with advanced hard-example criteria, consistently improving continual graph learning performance.

✅ **We emphasize that the PDF has been revised to include efficiency evaluations and component comparison experiments in the appendix, with appropriate references incorporated in the main text.**

Best regards,

The Authors

---

### Meta-Review · Area_Chair_LPXE · 2025-12-29

**Summary:**

This paper studies Non-Exemplar Continual Graph Learning (NECGL), where graph neural networks must learn a sequence of class-incremental tasks without storing raw data from previous tasks. The authors argue that conventional Prototype Replay (PR) suffers from feature drift as the encoder evolves, and empirically observe that Prototype Contrastive Learning (PCL) exhibits reduced drift. Motivated by this observation, the paper proposes Instance-Prototype Affinity Learning (IPAL), which builds on PCL. The paper further presents a KL-divergence–based theoretical argument under Gaussian assumptions to justify why PCL should incur less drift than PR. Empirical evaluations on four benchmark datasets show that IPAL outperforms several baselines in terms of average accuracy and forgetting.

## Reviewers’ Concerns Before Rebuttal

The reviewers raised three major categories of concerns:

### Validity and usefulness of the theoretical results
Several reviewers questioned the theoretical analysis, noting that it relies on strong assumptions, including Gaussian feature distributions, positive-definite covariances, and infinitesimal (first-order) parameter updates. The results were viewed as qualitative at best, with unclear applicability to practical deep GNN training regimes.

### Novelty of the proposed method
Reviewers expressed concern that the proposed framework appears largely incremental. The use of prototype-based modeling, feature/representation drift control, and regularization across tasks is well established in continual learning, including in graph continual learning. It was unclear what fundamentally new insight or capability is enabled by the specific Gaussian modeling or instance–prototype affinity formulation.

### Computational overhead and scalability
Multiple reviewers pointed out that the method introduces additional components (PageRank, covariance estimation, hard-sample mining) but initially lacked concrete runtime, memory, and scalability analysis, particularly on large graphs.

## Assessment After the Rebuttal

After the rebuttal, only the computational efficiency and scalability concern is convincingly addressed. The authors provided extensive runtime and memory evaluations, showing that PageRank and the additional modules introduce manageable overhead and are comparable to existing prototype-replay methods.

However, the core concerns regarding novelty and theory remain largely unresolved:

### Incremental nature of the method:
At its core, the proposed approach models feature drift using class-wise mean and variance, i.e., a Gaussian assumption, and penalizes drift relative to this model. Penalizing drift—either directly or indirectly—has been widely explored in continual learning, including graph continual learning. The paper does not clearly articulate what new capability or insight is enabled specifically by the Gaussian modeling assumption, beyond a convenient mathematical abstraction.

### Limited value of the theoretical analysis:
The theoretical results fundamentally rely on the same Gaussian modeling assumption that motivates the method. As a result, the theory does not independently validate or strengthen the approach; instead, it largely restates the modeling assumption in analytical form. Moreover, the proof itself is informal and handwaving, relying on first-order approximations and qualitative gradient arguments without rigorously establishing meaningful bounds or guarantees. Consequently, the theoretical section provides little additional value in justifying the method’s effectiveness.

Overall, while the rebuttal improves the paper’s completeness in terms of efficiency reporting, it does not substantially strengthen the conceptual or theoretical foundations of the work.

## Concerns Regarding the Experimental Study

The experimental evaluation also raises concerns. The comparisons are conducted primarily against general continual learning baselines, many of which are not tailored to graph neural networks. In particular, the reported performance of ER-GNN appears unusually low compared to results reported in the original ER-GNN paper and in established continual graph learning benchmarks. This casts doubt on whether the baselines are optimally tuned or fairly evaluated, and weakens the empirical evidence supporting the claimed improvements.

In summary, although the paper presents a coherent framework and the authors have addressed the computational efficiency concerns, significant issues remain regarding novelty, theoretical rigor, and experimental validation. The proposed method appears largely incremental relative to prior work, the theoretical analysis is weak and heavily assumption-dependent, and the experimental comparisons raise questions about fairness and representativeness.

Based on the remaining reviewer concerns and the issues outlined above, I do not believe the paper is ready for publication at this stage.

**Reviewer Concerns:**

See the summary above

**Reviewer Scores:**

I do not expect any reviewer would raise their score as the main concerns of the paper are not well-addressed by the rebuttal.

---

### Decision · Program_Chairs · 2026-01-26

Reject